# REV7 is required for processing AID initiated DNA lesions in activated B cells

Dingpeng Yang [1,2], Ying Sun[3], Jingjing Chen[4], Ying Zhang[1], Shuangshuang Fan [1], Min Huang[1,2], Xia Xie[1,2], Yanni Cai[1], Yafang Shang[1,2], Tuantuan Gui[5], Liming Sun [1,2], Jiazhi Hu [6], Junchao Dong[7], Leng-Siew Yeap[5], Xiaoming Wang [4], Wei Xiao[3] & Fei-Long Meng [1,2✉]

Activation-induced cytidine deaminase (AID) initiates both antibody class switch recombination (CSR) and somatic hypermutation (SHM) in antibody diversification. DNA double-strand break response (DSBR) factors promote rearrangement in CSR, while translesion synthesis (TLS) polymerases generate mutations in SHM. REV7, a component of TLS polymerase zeta, is also a downstream effector of 53BP1-RIF1 DSBR pathway. Here, we study the multi-functions of REV7 and find that REV7 is required for the B cell survival upon AID-deamination, which is independent of its roles in DSBR, G2/M transition or REV1-mediated TLS. The cell death in REV7-deficient activated B cells can be fully rescued by AID-deficiency in vivo. We further identify that REV7-depedent TLS across UNG-processed apurinic/apyrimidinic sites is required for cell survival upon AID/APOBEC deamination. This study dissects the multiple roles of Rev7 in antibody diversification, and discovers that TLS is not only required for sequence diversification but also B cell survival upon AID-initiated lesions.

[1] State Key Laboratory of Molecular Biology, Center for Excellence in Molecular Cell Science, Shanghai Institute of Biochemistry and Cell Biology, Chinese Academy of Sciences, Shanghai 200031, China. [2] University of Chinese Academy of Sciences, Beijing 100049, China. [3] College of Life Sciences, Capital Normal University, Beijing 100048, China. [4] Department of Immunology, State Key Laboratory of Reproductive Medicine, Nanjing Medical University, Nanjing, Jiangsu 211166, China. [5] Department of Immunology and Microbiology, Shanghai Institute of Immunology, Shanghai Jiao Tong University School of Medicine, Shanghai 200025, China. [6] The MOE Key Laboratory of Cell Proliferation and Differentiation, Genome Editing Research Center, School of Life Sciences, Peking-Tsinghua Center for Life Sciences, Peking University, Beijing 100871, China. [7] Department of Immunology, Zhongshan School of Medicine, Sun Yat-sen University, Guangzhou 510080, China. ✉email: Feilong.Meng@sibcb.ac.cn

Upon antigen stimulation, mature B cells can undergo antibody diversification processes including immunoglobulin heavy chain (IgH) class switch recombination (CSR) and variable (V) exon somatic hypermutation (SHM)[1]. Activation-induced cytidine deaminase (AID) initiates both CSR and SHM[2] by specific targeting to immunoglobulin loci and converting cytosine (C) to uracil (U)[3]. Various DNA repair pathways function downstream and channel the deamination products into double-strand breaks (DSBs), mutations, or small insertions/deletions (indels). Related DNA repair factors are critical for immune diversity and their deficiency could lead to primary immunodeficiency in human patients.

During CSR, deamination products are processed by base excision repair (BER) and mismatch repair (MMR) factors to generate DSBs at upstream Switch (S) region and downstream S regions[4]. The DSB activates DSB response (DSBR) factors including Ataxia telangiectasia mutated (ATM) and its substrates H2AX, 53BP1, etc. Eventually, non-homologous end joining (NHEJ) pathway juxtaposes the two S breaks and changes the antibody class from IgM to other isotypes[4]. In DSBR, 53BP1 is the key factor to limit DSB end resection and promote NHEJ[5]. 53BP1 is recruited to DSB site through a dual H4K20me2 and H2AK15ub histone marks[6], and its N-terminal domain further recruits PTIP and Rif1 to inhibit end resection upon phosphorylation[7]. Deletion of either factor in this pathway results in decreased CSR level in various mouse models[8–10].

SHM specifically happens at Ig V exons in germinal center (GC) B cells in vivo[11]. U can be recognized by uracil DNA glycosylase (UNG) and processed into an apurinic/apyrimidinic (AP) site, which can also be cut into an single-stranded DNA nick by apurinic/apyrimidinic-endonuclease[12]. The U-G mismatch can also be recognized by MMR factors and DNA strand containing mismatch will be cleaved to generate a single-nucleotide gap[12]. Different from the canonical BER or MMR process, SHM utilizes error-prone translesion DNA synthesis (TLS) to fill in the gap/nick/AP site[13] as a balance between error-free and error-prone DNA repair[14]. In this context, many TLS polymerases are involved in SHM. REV1 can add C to the opposite position of AP site to produce a C-to-G transversion[15]. DNA polymerase eta (POLH) and DNA polymerase zeta (POLZ complex, including REV7 and the catalytic subunit REV3L) can fill in the gap/nick to generate A/T mutations[16] or tandem mutations[17,18], respectively. The uracil processing by UNG and MutS homolog (MSH) proteins are the key steps, as U's can only be processed during DNA replication to generate C to T transition mutations in $Msh2^{-/-}Ung^{-/-}$ mice[19]. Although DSB was detected at a lower frequency at Ig V exon during SHM, DSBR- or NHEJ-deficient mice possess normal SHM levels, suggesting that DSBR or NHEJ is not required for mutation in SHM[8].

Rev7 was first identified in a genetic screening of UV mutagenesis in budding yeast[20] and the Rev7 protein was identified as a component of POLZ together with Rev3[21]. Later, Rev7 was found to be a HORMA domain (conserved domain found in budding yeast Hop1p, Rev7p, and MAD2 proteins) containing protein that can interact with many other proteins via a stereotypical safety-belt peptide interaction mechanism[22]. Besides Rev3[21], Rev7 interacts with Rev1[23], CDH1[24], and many others, supporting its multiple roles in DNA translesion synthesis, the anaphase-promoting complex/cyclosome (APC/C) inhibition[24], spindle assembly[25], etc. REV7-deficient human cells show UV[26], cisplatin, and irradiation (IR) hypersensitivity[27,28], suggesting the involvement of REV7 in different DNA repair pathways. REV7 was found to inhibit DNA resection at DSB and promote NHEJ in a 53BP1-RIF1-dependent pathway[29,30], and also identified as a Fanconi anemia gene from human patients[31]. Recently, a REV7-SHLD1/2/3 complex (Shieldin) was identified by many groups,

which functions as a downstream effector of 53BP1-RIF1 in promoting NHEJ through inhibition of DNA end resection[32–36].

Antibody gene diversification process starts with programmed DNA lesions and relies on many DNA repair pathways to generate diverse DNA sequences, which is an excellent in-vivo model to dissect physiological functions of DNA repair factors[1]. During CSR, DSB end joining can be quantitatively visualized by testing antibody switch frequency, whereas DSB end resection can be measured with IgH region junctions with high-throughput genome-wide translocation sequencing (HTGTS)[37,38]. With a high-throughput sequencing-based SHM assay and pipeline, mutation frequency and spectrum can by retrieved from more than 100 thousand mutated nucleotides to achieve statistical significance[39].

In this study, we generate a B-cell-specific Rev7-knockout mouse model and study CSR and SHM in Rev7-knockout B cells. We find that Rev7 is crucial for both CSR and SHM, and functions in these processes through different pathways. REV7 promotes CSR via the recently identified 53BP1-RIF1-Shieldin pathway, whereas REV7-REV3L are required for B-cell survival upon AID-initiated DNA lesions.

## Results

**REV7 deficiency leads to B-cell death during CSR.** To dissect REV7's multiple roles in antibody diversification, we generated a Rev7 floxed mouse model (Supplementary Fig. 1a) and bred it with Cd19cre mice[40]. Similar to a recent report[35], total splenic B-cell numbers were indistinguishable between REV7-deficient and control mice (Supplementary Fig. 1b). Splenic naive B cells were purified and stimulated with lipopolysaccharide (LPS) plus interleukin-4 (IL4) or LPS alone to induce CSR to IgG1 or IgG3 ex vivo (named as CSR-activated B cells). REV7 defieicncy led to defective CSR (Fig. 1a and Supplementary Fig. 1c, d) as previously shown in B cells[29,30,35], without affecting AID protein level, germline transcription of Ig constant genes (Supplementary Fig. 1e, f).

However, when we counted the live cell numbers after cytokine stimulation, we noticed a growth defect in CSR-activated REV7-deficient B cells but not in 53BP1 deficiency (Fig. 1b). Dramatic growth defect of REV7-deficient B cells was observed at Day 4 after cytokine stimulation (Supplementary Fig. 1g), which showed ~50% B-cell numbers of that in controls. Consistently, when the cells were analyzed by using flow cytometry, numbers of cell fraction in live population (gated by forward and side scatters) decreased significantly (Supplementary Fig. 1h). The extensive CSR events happened between Day 3 and four time points (Fig. 1a and Supplementary Fig. 1d) could contribute to the much severe cell death at Day 4 under the current culture condition. B cells in the gated live population display a slight defect in cell division revealed by a carboxyfluorescein diacetate succinimidyl ester (CFSE) cell-labeling assay (Supplementary Fig. 1i). To investigate the cause of cell death, we measured the cell apoptosis with Annexin V/propidium iodide dual staining and found REV7 deficiency results in increased apoptotic population (Fig. 1c and Supplementary Fig. 1j). Similar reduction of CSR-activated REV7-deficient B cells was observed in both cell number counting and cytometry assays upon LPS plus IL4 or LPS stimulation (Supplementary Fig. 1g, h), except less apoptotic REV7-deficient B-cell population was observed upon LPS plus IL4 stimulation, potentially reflecting a role of IL4-initiated anti-apoptotic single transduction pathway[41].

To identify the potential pathway ensuring B-cell survival upon CSR activation, we dissected REV7's multiple roles (Fig. 1d) during CSR using different assays. First, we performed HTGTS assay to analyze AID-break resections at S regions (Supplementary Fig. 2a)[37]. REV7 deficiency resulted in expanded resection of

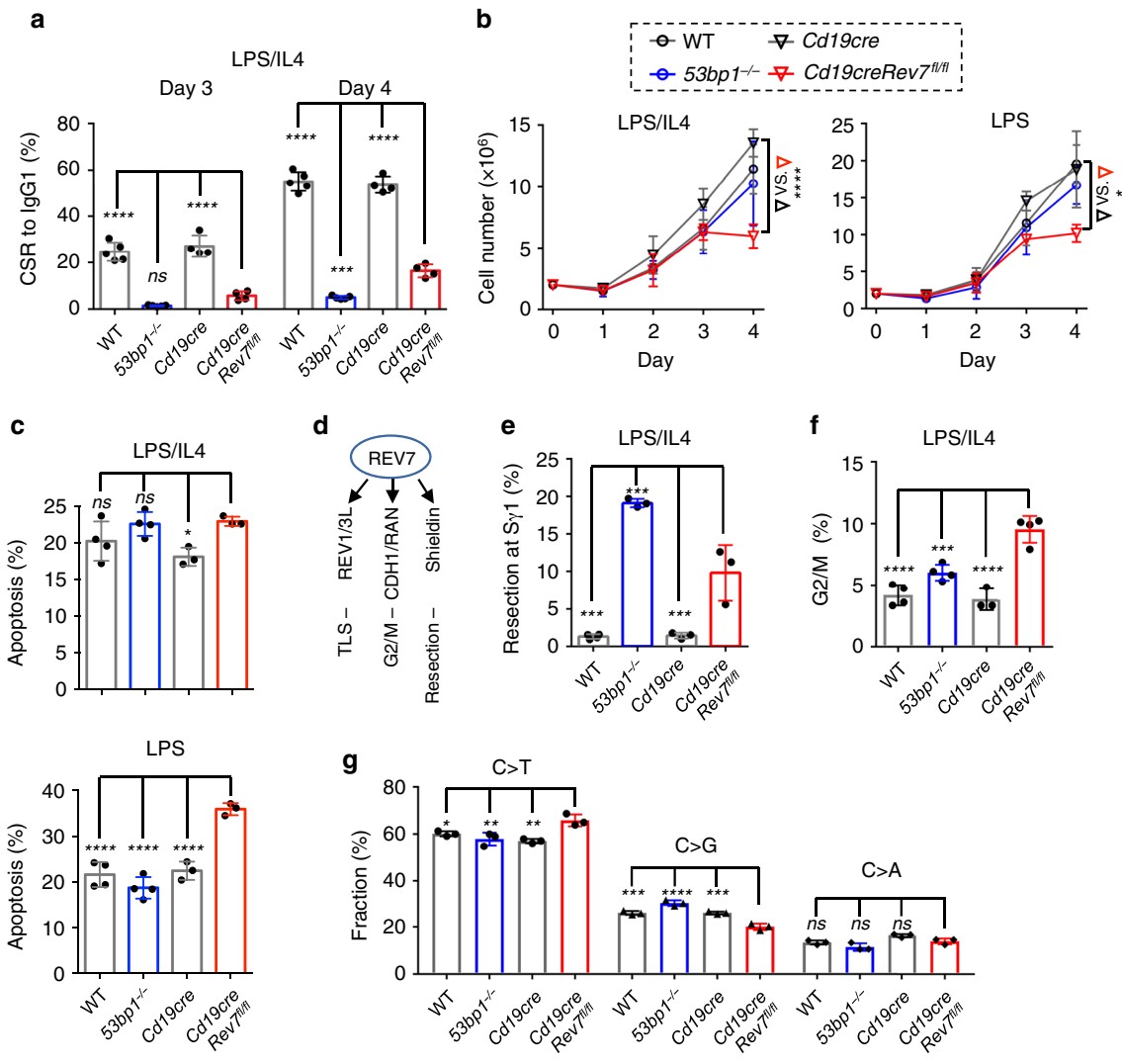

**Fig. 1 REV7 is required for B-cell viability during CSR. a** CSR levels to IgG1 after LPS/IL4 stimulation at Day 3 and 4. $n = 5$ for WT and $53bp1^{-/-}$; $n = 4$ for $Cd19cre$ and $Cd19cre\ Rev7^{fl/fl}$. **b** Growth curve of indicated B cells upon LPS/IL4 (left) or LPS (right) stimulation. Cell numbers at Day 4 of four sets of mice are subjected to statistical calculation. $n = 4$ for all genotypes. **c** Percentages of apoptotic cell (AnnexinV$^+$PI$^-$) at Day 4 after stimulating with LPS/IL4 (upper) or LPS (lower) are plotted. $n = 4$ for WT and $53bp1^{-/-}$; $n = 3$ for $Cd19cre$ and $Cd19cre\ Rev7^{fl/fl}$. **d** Multiple roles of REV7. **e** End resection of AID-initiated breaks at S$\gamma$1 is shown for indicated B cells. Resection ratio is defined as previously reported[37]. $n = 4$ for WT; $n = 3$ for $53bp1^{-/-}$, $Cd19cre$, and $Cd19cre\ Rev7^{fl/fl}$. **f** Percentage of cells in G2/M phase is plotted for the indicated genotypes. $n = 4$ for WT, $53bp1^{-/-}$, and $Cd19cre\ Rev7^{fl/fl}$; $n = 3$ for $Cd19cre$. **g** Mutation spectrum of C/G in the 5′ S$\mu$ region is shown for both strands of genomic DNA. $n = 3$ for all genotypes. $n$, independent mice. Data are represented as mean ± SD in **a**, **b**, **c**, **e**, **f**, and **g**. Two-tailed unpaired $t$-test was performed for **b** and one-way ANOVA followed by Dunnett's multiple comparisons test was performed for **a**, **c**, **e**, **f**, and **g**. Data from $Rev7$ knockout are compared with those from other genotypes. ****$p < 0.0001$, ***$p < 0.001$, **$p < 0.01$, *$p < 0.05$, ns: $p > 0.05$. $P$-values and sample sizes are provided in Supplementary Table 2. Source data are provided as a Source Data file.

S region DNA ends at a less severe level compared with that in 53BP1 deficiency (Fig. 1e and Supplementary Fig. 2b, c), suggesting DNA resection unlikely contributes to cell death in REV7 deficiency. Next, we examined cell cycle in the CSR-activated B cells (Supplementary Fig. 3) and found significant G2/M arrest in REV7 deficiency (Fig. 1f), which reflects REV7's role in G2/M transition[24,25].

Then, we analyzed mutations of the 5′ S$\mu$ region in CSR-activated B cell by high-throughput sequencing[39], to access its role in TLS-mediated mutation. Although DSBs are the major outcomes of AID deamination in CSR, mutations at S regions are frequently observed[42]. Mutations in the ~200 bp 5′ S$\mu$ amplicon were analyzed via a SHM pipeline[39]. In DSBR deficiencies including $53bp1$, $Atm$, and $Rev7$ deletion, fraction of reads with mutations was significantly decreased (Supplementary Fig. 4a). Further deletion of $Atm$ in 53BP1 deficiency partially rescued

expanded end resection but did not change the mutation frequency (Supplementary Fig. 4a–c) and deletion of DSBR genes in $Ung^{-/-}Msh2^{-/-}$ CH12F3 cells had no effect on S region mutation frequency (Supplementary Fig. 4d–f), reflecting that many AID lesions were subjected to breakage and excluded from the amplicon-seq in DSBR deletion cells or some of these genes are required for converting the AID lesion into DSBs. In CSR, the downstream DNA repair pathways are different from SHM in generation mutation outcome[43]. However, the 5′ S$\mu$ amplicon-seq allowed the analysis of mutation spectrum on C/G in these mutants, which could be an assay to study TLS. In this context, we found that C > G transversion was significantly decreased in REV7 deficiency but not in 53BP1 deficiency (Fig. 1g), correlating with the REV1/REV7-dependent C > G during TLS[15].

Thus, many aspects of REV7 functions can be visualized during CSR (Fig. 1d), which offers an experimental model to dissect its

multiple roles including the unexpected cell death in REV7-deficient CSR-activated B cells.

**REV7 and REV3L protect activated CH12 cells from cell death.**
To study the molecular basis for the cell death in CSR-activated REV7-deficient B cells, we made a panel of knockouts using CRISPR/Cas9 in B-lineaged CH12F3 cells (Supplementary Fig. 5a), which can undergo CSR to IgA upon anti-CD40/IL4/TGF-β (CIT) stimulation[44]. The gene knockouts were genotyped by PCR from genomic DNA and western blotting with whole-cell lysate (Supplementary Fig. 5a), except for *Shld3* whose antibody is not commercially available so far, and the knockout was verified by using reverse transcription quantitative real-time PCR (RT-qPCR) (Supplementary Fig. 5a). The knockout clones were further confirmed by their known functions in repairing γ-IR-, cisplatin-, and ultraviolet (UVC, wavelength 254 nm)-caused DNA damages (Fig. 2a). In this context, 53BP1-RIF1-Shieldin pathway and *Rev3l* gene knockouts were sensitive to IR-generated DSBs (Fig. 2a, upper), whereas *Rev1*, *Rev3l*, and *Rev7* knockouts were sensitive to UV-generated crosslinks and cisplatin-caused DNA interstrand crosslinks (Fig. 2a). As REV1 protein can function as a scaffold for other TLS polymerases besides its catalytic activity[23,45], we also generated a *Rev1*$^{Δ9}$ cells expressing a REV1 protein without catalytic motif (in-frame Exon 9 deletion) along with *Rev1*$^{−/−}$ cells (out-frame Exon 10 deletion with no protein detected) (Supplementary Fig. 5a), and the corresponding cells showed similar sensitivity to DNA damages as previously reported[26–28,45].

Multiple independent knockout clones of each genotype were verified and stimulated with CIT for CSR. Consistent with previous reports, deletion of 53BP1-RIF1-Shieldin pathway genes led to decreased CSR and double knockouts of *Rev7* with 53BP1-RIF1-Shieldin pathway genes resulted in similar decreased CSR levels as those in single knockouts (Fig. 2b and Supplementary Fig. 5b). In *Rev1*$^{−/−}$ or *Rev1*$^{Δ9}$ cells, CSR level was not affected (Fig. 2b), similar to that reported in *Rev1*-knockout or catalytically inactivate *Rev1* mutant mouse models[45,46] but different from another report[47], whereas in *Rev3l* deletion cells, CSR was decreased (Fig. 2b) as previously reported[18,48].

We then examined end resection of S region breaks with HTGTS method in all mutant cell lines. Rearrangement was cloned from a CRISPR/Cas9-generated bait break at the Iγ3 region to avoid the interference of the nonproductive *IgH* allele in CH12 cells, which already undergoes IgM-IgA switching[37,38], and rearrangements between Iγ3$^{Cas9}$ and Sμ/α$^{AID}$ were analyzed (Supplementary Fig. 6a). Expanded S region resection was observed in all 53BP1-RIF1-Shieldin-deficient cell lines but not in REV1 or REV3L deficiency (Fig. 2c and Supplementary Fig. 6b). However, significantly decreased C > G transversion of 5′ Sμ mutations was observed in REV1, REV3L, and REV7 deficiencies (Fig. 2d and Supplementary Fig. 6c). We also examined the cell cycle of these knockout CH12 cell lines. Similar to previous observation[49], RIF1 deficiency results in a significant G2/M arrest (Supplementary Fig. 7). Similar G2/M arrest phenotype was observed in both REV7 and REV3L deficiencies regardless of CIT stimulation or not (Supplementary Fig. 7). The panel of CH12F3 knockouts helps to dissect the multiple roles of REV7 in DSBR and TLS, in which context end resection is counteracted by 53BP1-RIF1-Shieldin and C > G transversion is mediated by REV1/3L/7 TLS polymerases.

We then examined cell growth in these genotypes and found that growth defect was only observed in *Rev7* or *Rev3l*-knockout cells upon CIT stimulation (Fig. 2e). In apoptosis assay, *Rev7* or *Rev3l*-knockout cells showed increased apoptotic population in either CIT stimulation conditions and a slight but significant increasing in non-CIT condition (Fig. 2f). Combining the results of end resection, C > G transversion, G2/M arrest, and cell viability in these gene knockout cell lines, we found the REV7 is required for cell survival upon AID lesions in CH12 cells, for which the phenotype is also observed in REV3L deficiency.

**Residues in HORMA domain are crucial for B-cell survival.**
Key residues on REV7 protein responsible for its interactions to cofactors were clearly mapped (Fig. 3a). Thus, we used these mutants to dissect REV7's roles in CSR. REV7 mutants were overexpressed in REV7-deficient B cells with a retroviral vector with wild-type (WT) REV7 and empty vector (EV) as controls. All cellular defects in REV7-deficient B cells can be fully rescued by overexpressing WT REV7 (Fig. 3b, c and Supplementary Fig. 8a).

Decreased CSR and increased end resection were observed in mutants losing interaction of other Shieldin subunits, including Y63A (abolished interaction with SHLD3[35]) and K129A (abolished interaction with SHLD2[35]) (Fig. 3b, c and Supplementary Fig. 8b, c). The L186A mutant, which fails to interact with REV1[50], fully rescued the CSR defect and expanded S region resection (Fig. 3b, c and Supplementary Fig. 8b, c), but showed increased C > G transversion frequency (Fig. 3d and Supplementary Fig. 9), suggesting an interaction between REV1 and REV7 is required for proper C > G transversion. In the HORMA domain, both Y63 and W171 contribute to REV7–REV3L interaction[50], whereas only Y63 is the major contributor for in-vivo REV7–SHLD3 interaction[35,51]. The separation-of-function mutant W171A showed normal end-resection level and increased C > G mutation (Fig. 3c, d, and Supplementary Figs. 8b, c and 9), correlating with the observation that REV7–REV3L interaction affects REV1 function through the REV7–REV3L–REV1 complex[50]. In this context, the Rev7-Y63AW171A mutation[50] completely abolished its functions in CSR similar to the EV control (Fig. 3b and Supplementary Fig. 8a).

Complementation in *Rev7*-deleted CH12F3 cells yield same results as in CSR-activated primary B cells (Fig. 3e). The cell growth and viability were examined in mutated CH12F3 cells and only Y63AW171A and V85E showed growth defect and increased cell death upon CIT stimulation (Fig. 3f, g). The W171A mutant did not affect cell death (Fig. 3f, g), probably resulting from its remaining interaction with REV3L[50]. Combining the cellular outputs of various REV7 mutants (Fig. 3), we conclude that the intact of REV7 HORMA domain is required for its function in promoting B-cell survival, potentially by its interaction with REV3L via the safety-belt model.

**AID-initiated lesions cause cell death in REV7 deficiency.** The much severe cell death observed upon CIT stimulation in REV7-deficient CH12F3 cells (Fig. 2f) indicated the B-cell activation could be the cause of cell death. The increased cell death at Day 4 vs. Day 3 after stimulation (Fig. 1b) prompted us to examine the Cre efficiency at the two time points (Fig. 4a). We designed a quantitative PCR assay to access the knockout efficiency. At Day 4 after stimulation, the *Rev7* floxed allele was significantly increased indicating a counter-selection of deletion alleles of *Cd19creRev7*$^{fl/fl}$ B cells (Fig. 4a). Thus, the *Cd19creRev7*$^{fl/fl}$ mice were bred with *Aicda*$^{−/−}$ mice to generate REV7 and AID double-deficient B cells. In the absence of AID, CSR was completely abolished (Supplementary Fig. 10a). Surprisingly, the growth defect and cell death of REV7 deficiency were fully rescued (Fig. 4b, c and Supplementary Fig. 10b). Accordingly, no significant change of deletion/floxed ratio was observed between

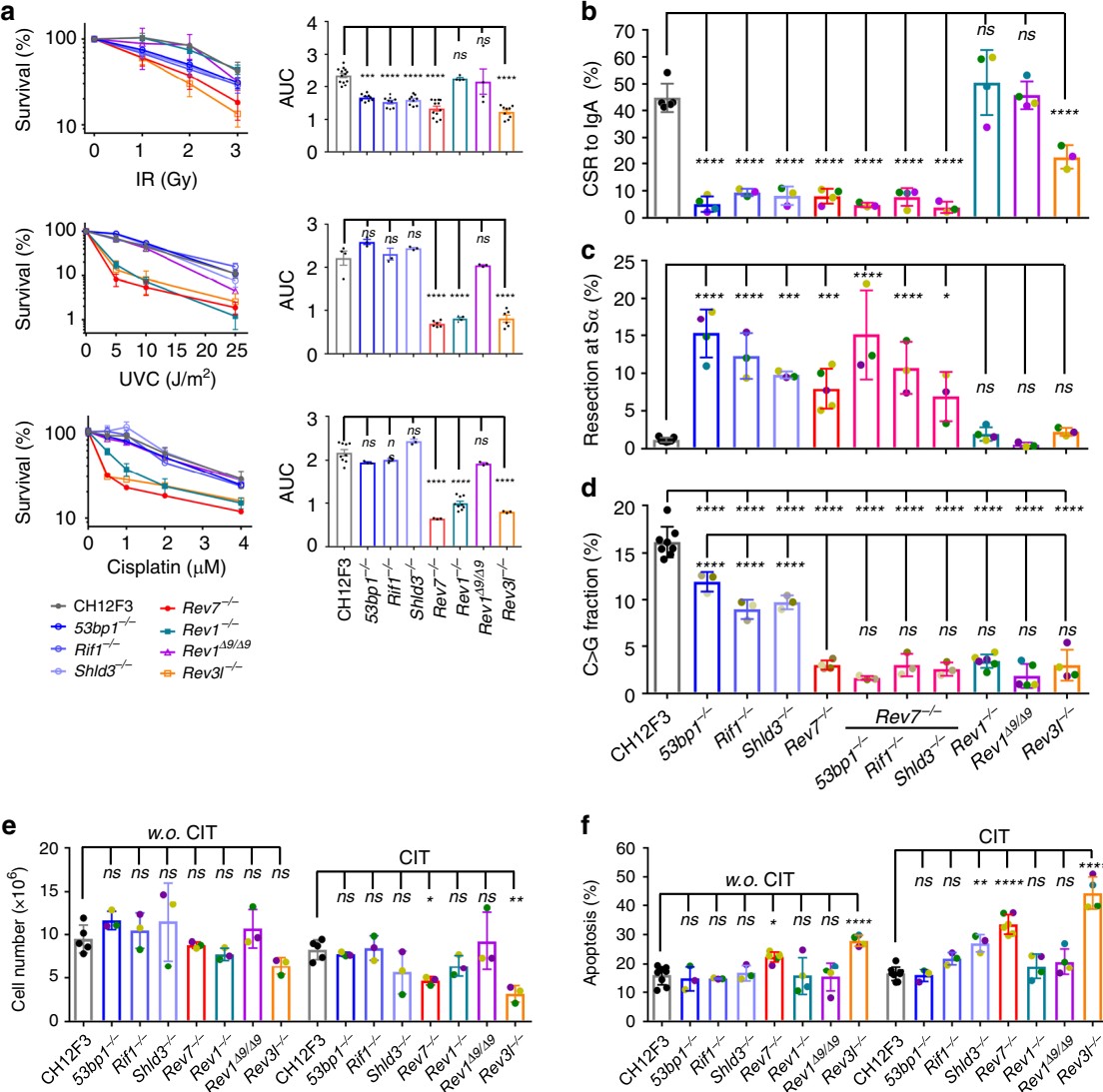

**Fig. 2 REV7 and REV3L protect activated CH12 cells from cell death. a** Survival curves of indicated CH12F3 cell lines upon γ-irradiation (IR), UVC, and cisplatin treatment are plotted as means ± SD at left. Area-under-curve (AUC) are calculated and means ± SEM are compared at the right. In IR treatment, $n = 13$ for parental CH12F3 cells; $n = 12$ for $Rev7^{-/-}$; $n = 4$ for $Rev1^{-/-}$ and $Rev1^{\Delta9/\Delta9}$; $n = 9$ for the other genotypes. In UVC treatment, $n = 4$ for parental CH12F3 cells and $Rev1^{-/-}$; $n = 6$ for $Rev7^{-/-}$ and $Rev3l^{-/-}$. $n = 3$ for the other genotypes. In cisplatin treatment, $n = 9$ for parental CH12F3 cells; $n = 9$ for $Rev1^{-/-}$; $n = 3$ for the other genotypes. CSR rate from IgM to IgA (**b**), end resection level in Sα region (**c**), and percentage of C > G transversion in 5'Sμ region (**d**) are shown for the indicated cells. Cell numbers (**e**) and percentage of apoptotic population (**f**) with/without cytokine stimulation (CIT, w.o.CIT) at Day 3 are showed. Colored points indicate individual knockout clones. In **b**, $n = 5$ for parental CH12F3 cells; $n = 4$ for $Rev7^{-/-}$, $53bp1^{-/-}$, $Rev7^{-/-}Rif1^{-/-}$, $Rev1^{-/-}$, and $Rev1^{\Delta9/\Delta9}$; $n = 3$ for the other genotypes. In **c**, $n = 7$ for parental CH12F3 cells; $n = 5$ for $Rev7^{-/-}$; $n = 4$ for $53bp1^{-/-}$ and $Rev1^{-/-}$; $n = 3$ for the other genotypes. In **d**, $n = 8$ for parental CH12F3 cells; $n = 4$ for $Rev7^{-/-}$ and $Rev3l^{-/-}$; $n = 6$ for $Rev1^{-/-}$; $n = 5$ for $Rev1^{\Delta9/\Delta9}$, $n = 3$ for the other genotypes. In **e**, $n = 5$ for parental CH12F3 cells; $n = 3$ for the other genotypes. In **f**, $n = 8$ for parental CH12F3 cells; $n = 6$ for $Rev7^{-/-}$; $n = 4$ for $Rev1^{-/-}$, $Rev1^{\Delta9/\Delta9}$, and $Rev3l^{-/-}$; $n = 3$ for $53bp1^{-/-}$, $Rif1^{-/-}$, and $Shld3^{-/-}$. Three or more independent clones for each genotype were assayed and $n$ represents independent experiments. Data are represented as mean ± SD; one-way ANOVA followed by Dunnett's multiple comparisons test was performed for all panels. Data in knockouts are compared with those in parental CH12F3 cells for all panels. In **d**, an extra comparison is shown by using $Rev7^{-/-}$ as the reference group to highlight the difference between DSBR deficiencies and TLS deficiencies. ****$p < 0.0001$, ***$p < 0.001$, **$p < 0.01$, *$p < 0.05$, ns: $p > 0.05$. Source data are provided as a Source Data file.

Day 3 and 4 after stimulation (Fig. 4d). However, the G2/M arrest in REV7 deficiency could not be rescued by *Aicda* deletion (Fig. 4e and Supplementary Fig. 10c). The G2/M fractions of CSR-activated REV7 single-deficient and REV7/AID double-deficient B cells were at a comparable level (Figs. 1f and 4e), suggesting the G2/M phase arrest is not a major contributor of the higher apoptosis rate in REV7-deficient cells. We conclude that AID-initiated DNA lesions are the cause of cell death in REV7 deficiency.

**AID-initiated lesions lead to dysfunctional GC in REV7 deficiency.** SHM of *Ig V* exons is required for antibody affinity maturation and also initiated by AID in GC B cells[12]. Cell death caused by AID-initiated DNA lesions in REV7 or REV3L deficiency during CSR indicated similar mechanism could happen in vivo in SHM. The GC B-cell numbers in the spleen or Peyer's patch of SRBC (sheep red blood cell)-immunized REV7-deficient mice were significantly decreased (Fig. 5a, b). The size of GC decreased in REV7-deficient spleens (Fig. 5c) and the floxed allele

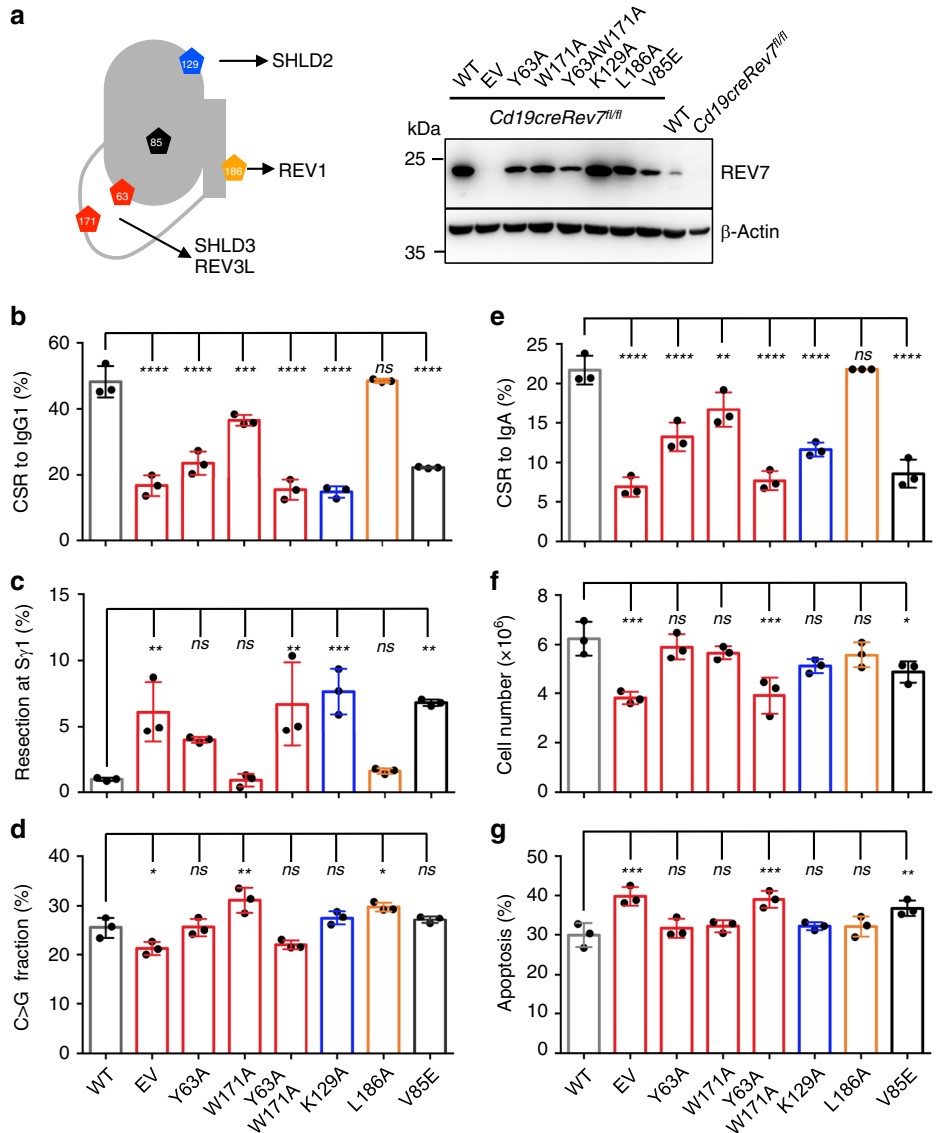

**Fig. 3 REV7 HORMA domain is required for B cell survival upon AID expression. a** Schematic illustration of critical residues on REV7 protein (left). Representative western blot of REV7 mutant proteins in B cells from three replicates are shown at right. EV: empty vector control. CSR levels to IgG1 (**b**), end resection level in Sγ1 region (**c**), and percentage of C > G transversion in 5′Sμ region (**d**) after LPS/IL4 stimulation at Day 4 of REV7-complementated primary B cells. CSR levels to IgA (**e**), cell numbers (**f**), and percentage of apoptotic population (**g**) of indicated CH12F3 cells with cytokine stimulation (CIT) at Day 3 are shown. $n = 3$ for each genotype in **b**–**g** and $n$ represents independent experiments. Data are represented as mean ± SD, one-way ANOVA followed by Dunnett's multiple comparisons test was performed for **b**–**g**. Data from cells complemented with WT REV7 protein are used as reference group in comparison. ****$p < 0.0001$, ***$p < 0.001$, **$p < 0.01$, *$p < 0.05$, ns: $p > 0.05$. P-values and sample sizes are provided in Supplementary Table 2. Source data are provided as a Source Data file.

significantly increased in the remaining GC B cells (Fig. 5d), indicating the counter-selection of *Rev7*-knockout B cells in GC. The counter-selection of REV7-knockout GC B cells suggested that AID-initiated DNA lesions are very toxic in the absence of REV7 in vivo and most survived GC B cells are either REV7-proficient or with low AID expression.

We thus examined mutation profiles of J$_H$4 intron and Jκ5 intron[52,53] in the REV7-deficient GC B cells. We found the mutation frequency was significantly decreased in the *Cd19creRev7fl/fl* GC B cells (Supplementary Fig. 11a, b). Consistent with a previous report[8], 53BP1 deficiency does not affect SHM frequency in the *Ig V* exon regions (Supplementary Fig. 11a, b), suggesting the decreased mutation frequency in REV7 deficiency is unlikely caused by its role in DSBR. We then checked the mutation spectrum of J$_H$4 and Jκ5 introns, and found that the

overall mutation spectrum in REV7 deficiency was similar to the spectrum in WT (Supplementary Fig. 11c–f). The counter-selection of REV7 deficiency and differential DNA repair pathways may contribute to the mild effect of mutation spectrum in GC B cells.

Similar to the phenotype observed in CSR-activated B cells (Fig. 4b), the decreased GC B-cell numbers can be fully rescued by the *Aicda* deletion in the spleen or Peyer's patch (Fig. 5e, f). Thus, REV7 is required for GC B-cell survival upon AID-initiated DNA lesions.

**Unrepaired AP site leads to B-cell death.** AID-initiated deamination products U's are processed through BER or MMR factors into AP site or nicks, which can be further channeled into DSB during CSR. The REV7/REV3L-dependent and DSBR-

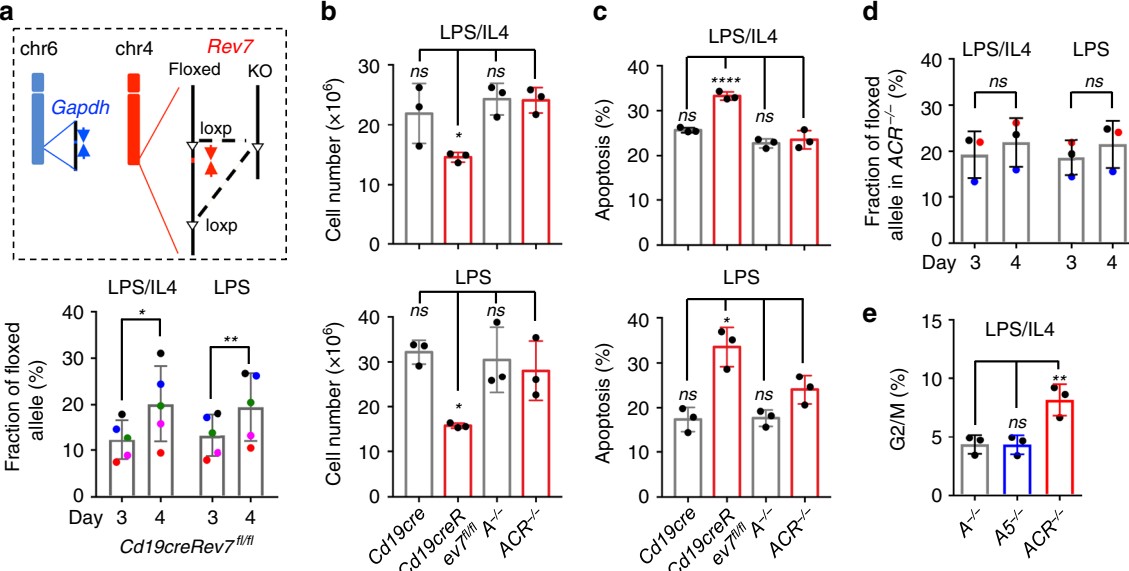

**Fig. 4 AID deficiency rescues the cell death of REV7-deficient B cells in CSR. a** Assessment of *Rev7* deletion efficiency. Schematic illustration of genotyping strategy of *Rev7*-floxed allele (upper). Q-PCR primers are marks as arrows. A locus in *Gapdh* (chr6) was assayed as the loading control (blue arrows). Percentage of flowed allele are shown for *Cd19creRev7^fl/fl* B cells at Day 3 and 4 after stimulation (lower). Colored dot indicates data from same mouse. *n* = 5 for each genotype, *n* represents independent mice. Cell numbers (**b**) and percentage of apoptotic population (**c**) upon LPS/IL4 (upper) or LPS (lower) are plotted. **d** Genotyping of indicated CSR-activated B cells. **e** Percentage of cells in G2/M phase is plotted for the indicated genotypes. $A^{-/-}$: *Aicda^{-/-}*, $A5^{-/-}$: *Aicda^{-/-} 53bp1^{-/-}*, $ACR^{-/-}$: *Aicda^{-/-} Cd19creRev7^{fl/fl}*. *n* = 3 for each genotype in **b**–**e** and *n* represents independent mice. Data are represented as mean ± SD. Two-tail paired *t*-test was performed for **a** and **d**, and one-way ANOVA followed by Dunnett's multiple comparisons test was performed for **b**, **c**, and **e**. $ACR^{-/-}$ mice are used as reference group in comparison for **b**, **c**, and **e**. \*\**p* < 0.01, \**p* < 0.05, ns: *p* > 0.05. *P*-values and sample sizes are provided in Supplementary Table 2. Source data are provided as a Source Data file.

independent cell survival in activated B cells indicated intermediate products but not DSBs are the cause of cell death. Thus, we examined the cell growth in UNG and MSH2 double deficiency, in which background the majority of U's are processed by DNA replication machinery to generate C > T transition mutation[19]. Deletion of *Rev7* in UNG and MSH2 double deficiency did not lead to cell growth defect or increased apoptotic population (Fig. 6), suggesting U's were not the cause of cell death. Thus, we examined REV7-deficient B cell death in the absence of UNG or MSH2, we found *Ung^{-/-}Rev7^{-/-}* cells were no longer sensitive to AID deamination, whereas *Msh2^{-/-}Rev7^{-/-}* cells recaptured the increased cell death upon AID expression (Fig. 6). Thus, the unrepaired AP site generated by UNG glycosylase is the major cause of cell death in absence of REV7.

## Discussion

Here we report that multiple roles of REV7 are required to process AID-initiated DNA lesions in CSR-activated B cells ex vivo and GC B cells in vivo. In AID-initiated antibody diversification processes, REV7-REV3L replicates across the AP sites to ensure the cell proliferation, REV1-REV7-REV3L generates C > G transversion and REV7-SHLD1/2/3 functions downstream of 53BP1-RIF1 to inhibit resection of DSB ends.

The differential outcomes of AID activity in REV7-deficient cells can be dissected with our panels of gene knockout cell lines and REV7 mutants. During CSR, we found the DSBs at S regions undergo expanded end resection in REV7-deficient CSR-activated B cells, consistent with its role in Shieldin complex[32–36]. By examining the cell cycle, we found G2/M arrest in REV7-deficient cells resulted from REV7's role in G2/M transition[24,25]. We also noticed a decreased C > G transversion in the remnant 5′Sμ sequence, which is mainly contributed by the defective TLS of REV7 deficiency. During SHM, the mutation frequency at profiles of J_H4 and Jκ5 introns was decreased with no significant change

in mutation spectrum in REV7-deficient GC B cells. The absence of mutation spectrum change could be contributed by the counter-selection of AID expression REV7-deficient GC B cells, relative lower mutation rates in the *Ig V* introns, intrinsic sequence difference, or different downstream processing pathways. For example, in mutation spectrum analysis of 5′Sμ in CSR-activated B cells, no AT-spreading is observed, implicating a lack of POLH function in those B cells[16]. Although the mild changes of C > G transition could be masked by other error-prone polymerases, e.g., POLH during SHM in GC B cells. This hypothesis is supported by the previous observation by Saribasak et al.[17] that POLZ's function in SHM is masked by POLH. Mammalian S regions contain long repetitive GC-rich sequences[54] and the sequence intrinsic feature might also contribute to the different mutation profiles of *V* sequence and *S* sequence. In this context, we checked the mutation profile of VDJ exon in REV7-deficient CH12F3 cells (Supplementary Fig. 12). REV7 deficiency lead to a similar mutation profile, i.e., decreased mutation frequency (Supplementary Fig. 12a, b) and no significant change of mutation spectrum (Supplementary Fig. 12c).

Besides the well-characterized AID outcomes, we discovered an unexpected role of TLS in maintaining B-cell proliferation upon AID-initiated AP sites. The AID-dependent cell death of activated REV7-deficient B cells is independent of its role in G2/M arrest, as AID deficiency fully rescues the cell proliferation but not G2/M arrest. In mammalian cells, AP sites are replicated by TLS polymerases through a two-step process including insertion and extension[55]. In mammalian cells, many error-prone DNA polymerases including REV1 are involved in the insertion step, whereas POLZ (containing REV7-REV3L and other cofactors) is the major extender[56,57]. The AID-generated AP sites are very toxic and can result in cell death or depletion in GCs. In this context, POLZ not only diversifies DNA sequence but also ensures B cell proliferation. The AID-dependent cell death of

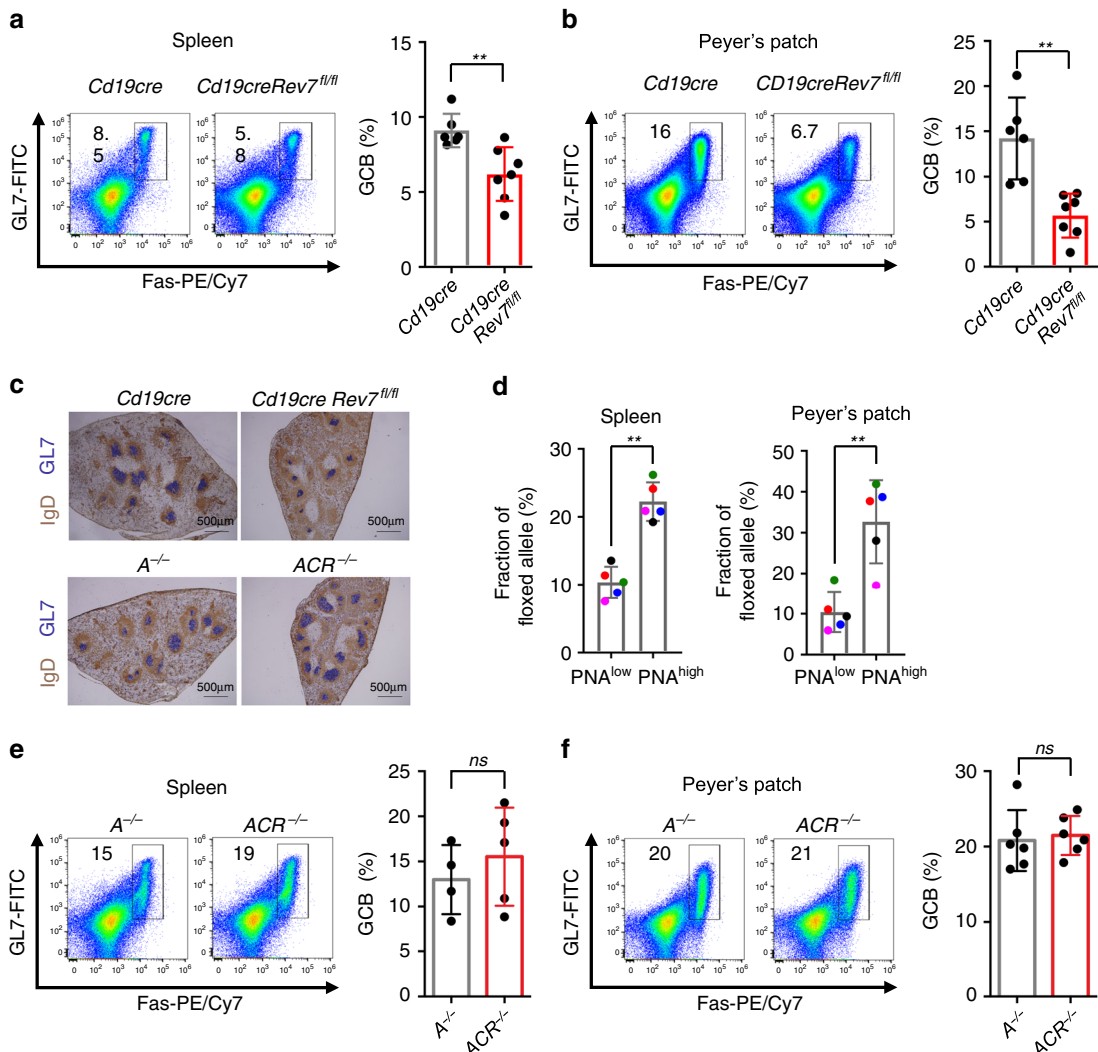

**Fig. 5 AID-dependent depletion of REV7-deficient GC B cells.** Fractions of GC B cells (GL7⁺Fas⁺) from spleens (**a**) and Peyer's patches (**b**) are showed with examplied flow cytometry plots and summary bar graphs. Each dot indicate result from an individual mouse. Six *Cd19cre* and seven *Cd19creRev7ᶠˡ/ᶠˡ* mice were assayed for **a** and **b**. **c** Immunohistochemistry of spleens of indicated genotype. Representative images from three sets of mice are showed. Scale bar, 500 μm, is indicated as a black line on each picture. **d** Genotyping of indicated B (B220⁺) cell populations (PNAˡᵒʷ: naive B cell, PNAʰⁱᵍʰ: GC B cell) in the spleen and Peyer's patch. *n* = 5 mice. AID deficiency fully rescued the GC B-cell depletion in spleens (**e**) and Peyer's patches (**f**) of Rev7-deficient mice. Four *Cd19cre* and five *Cd19creRev7ᶠˡ/ᶠˡ* mice were assayed for **e**. Six *Cd19cre* and *Cd19creRev7ᶠˡ/ᶠˡ* mice were assayed for **f**. Data are represented as mean ± SD. Two-tailed unpaired *t*-test was performed for **a**, **b**, **e**, and **f**; two-tailed paired *t*-test was performed for **d**. \*\**p* < 0.01, ns: *p* > 0.05. *P*-values and sample sizes are provided in Supplementary Table 2. Source data are provided as a Source Data file.

REV7 deficiency could happen quickly, as assessment of cell proliferation using CellTrace dye cannot reflect the defect as described in previous reported *Rev7* conditional knockout (*Mb1cre*) mouse model[35]. This uncovered function of REV7-REV3L also could help to understand the previous revealed CSR/SHM phenotype in *Rev3l* mutant mouse models[17,18,48,58]. REV3L is responsible for generation of tandem mutations demonstrated by using *Polh⁻/⁻Rev3l⁺/⁻*[17] or *Rev3l-hypermutant*[18] mouse models. We did not observe changes of tandem mutation frequency in our conditional *Rev7*-knockout mouse model, probably because the counter-selection of *Rev7*-knockout GC B cells, the redundant role of POLH, or separation of subunit functions within the POLZ complex[59].

AID/APOBEC cytidine deaminases are widely used in base editors[60]. Similar to AID-initiated antibody diversification, cytidine base editors could generate AP site that is crucial for cell survival in REV7 deficiency. In this context, AID or APOBEC3A (A3A) was ectopically expressed in B cells (Supplementary Fig. 13a), which was much more toxic to REV7-deficient cells (Supplementary Fig. 13b). Although UNG inhibitor UGI is usually included in the BE tools[60], the incomplete inhibition could lead to death of edited cells. In this context, base-editing tools generate a burden of DNA damages in certain genetic background, suggesting that extra caution should be taken when choosing genome editing tool in gene therapy of TLS mutant patients. On the other hand, more sophisticated base-editing tools can be designed based on the revealed downstream repair mechanism.

## Methods

**Mice.** *Rev7* floxed mouse line was constructed by this study. *Cd19cre*[40], *53bp1⁻/⁻*[9], *Atm⁻/⁻*[61], and *Aicda⁻/⁻*[62] mouse lines have been described previously. All mice were co-housed under specific pathogen-free conditions in sterile isolated cages at the animal core facility of Shanghai Institute of Biochemistry and Cell Biology. Mice were kept at maximum of 5 mice/cage at ambient temperature of 20–22 °C, 40–60% humidity under 8 am to 6 pm light cycle, 6 pm to 8 am dark cycle. All animal experiments were performed under protocols approved by the

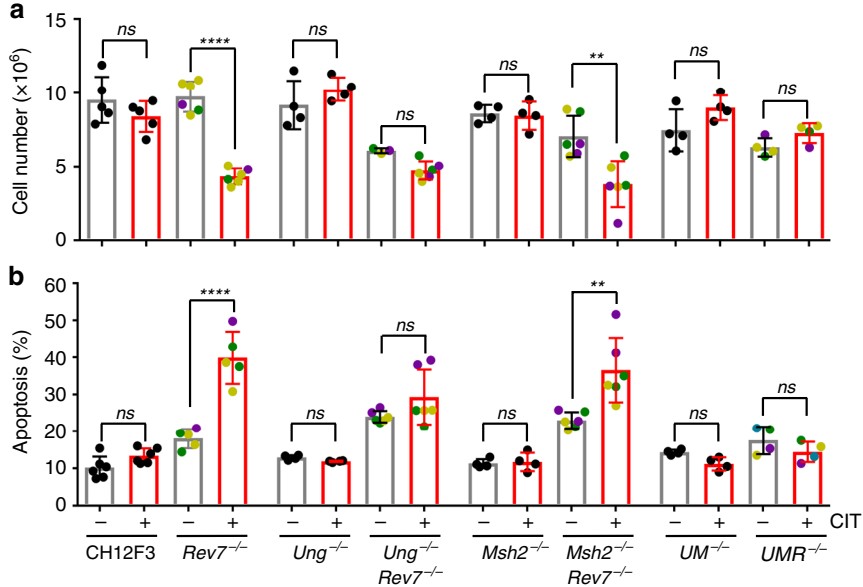

**Fig. 6 B-cell death depends on UNG-processed AID lesions.** Cell numbers (**a**) and percentage of apoptotic population (**b**) with(+)/without(−) cytokine stimulation (CIT) at Day 3 are showed for indicated genotypes. $UM^{-/-}$: $Ung^{-/-}Msh2^{-/-}$, $UMR^{-/-}$: $Ung^{-/-}Msh2^{-/-}Rev7^{-/-}$. Colored dots indicate individual clones. In **a**, $n = 5$ for parental CH12F3 cells; n = 6 for $Rev7^{-/-}$, $Ung^{-/-}Rev7^{-/-}$ (with CIT) and $Msh2^{-/-}Rev7^{-/-}$; n = 3 for $Ung^{-/-}Rev7^{-/-}$ (without CIT); n = 4 for the other genotypes. In **b**, $n = 6$ for parental CH12F3 cells, $Ung^{-/-}Rev7^{-/-}$ and $Msh2^{-/-}Rev7^{-/-}$; n = 5 for $Rev7^{-/-}$; n = 4 for the other genotypes. Three independent clones for each genotype were assayed, and n represents independent experiments. Data are represented as mean ± SD. One-way ANOVA followed by Dunnett's multiple comparisons test was performed for all panels. ****$p < 0.0001$, ***$p < 0.001$, **$p < 0.01$, ns: $p > 0.05$. P-values and sample sizes are provided in Supplementary Table 2. Source data are provided as a Source Data file.

Institutional Animal Care and Use Committee of Shanghai Institute of Biochemistry and Cell Biology.

**Cell lines**. Parental B-lineaged CH12F3 cell line[44] has been described previously. CH12F3 and its derived isogenic cells were cultured with RPMI1640 (10-040-CV, Corning), β-Mercaptoethanol (M6250, Sigma), and Penicillin–Streptomycin–Glutamine (10378016, Gibco, ThermoFisher Scientific), plus 10% fetal bovine serum (FBS) (FSP500, ExCell Bio). HEK293T cells were cultured with Dulbeccos modified Eagle's medium (10-013-CV, Corning), Penicillin–Streptomycin–Glutamine (10378016; ThermoFisher Scientific), plus 10% FBS (FSP500, ExCell Bio).

**Plasmids**. The pX330-U6-Chimeric_BB-CBh-hSpCas9 plasmid was obtained from Addgene (#42230). The coding sequence of Rev7, Aicda, and APOBEC3A was in-vitro synthesized and Rev7 mutants were obtained with site-directed mutagenesis, which were cloned into a retrovirus vector.

**Antibodies**. Antibodies for ATM (2873S; Cell Signaling; 1:1000), 53BP1 (NB100-304; NOVUSBIO; 1:1000), RIF1 (ab1213422; Abcam; 1:500), REV7 (A9861; Abclonal; 1:1000), REV1 (sc-393022; Santa Cruz; 1:1000), REV3L (GTX17515; Gene Tex; 1:1000), AID (A16217; Abclonal; 1:1000), MSH2 (ab227941; Abcam; 1:1000), β-actin (AC028; Abclonal; 1:10,000), FLAG (F1804, Sigma; 1:1000), β-Tubulin (A01030HRP; Abbkine; 1:10,000), glyceraldehyde 3-phosphate dehydrogenase (AB2000; Abways; 1:20,000), and Rabbit TrueBlot (18-8816-33; Ebioscience; 1:1000) were used in western blotting. PE-conjugated anti-mouse IgA (12-4204-83; Ebioscience; 1:200), APC-conjugated anti-mouse IgM (1020-11S; Southern biotech; 1:200), APC-conjugated anti-mouse B220 (553092, BD; 1:200), fluorescein isothiocyanate (FITC)-conjugated anti-mouse IgG1 (553443; BD; 1:200), FITC-conjugated anti-mouse IgG3 (553403; BD; 1:200), APC-eFluor780-conjugated anti-mouse B220 (47-0452-82, Invitrogen; 1:200), FITC-conjugated anti-mouse GL7 (144604, BioLegend; 1:200), PE-Cy7-conjugated anti-mouse CD95 (557653, BD; 1:200), and Fluorescein-labeled Peanut Agglutinin (FL-1071, Vector Laboratories; 1:500) were used in the fluorescence-activated cell sorting analysis.

**Primary B-cell culture and CSR assay**. Splenic naive B cells were purified with a mouse B-cell isolation kit from Stemcell (#19854) and the purified naive B cells were cultured at a density of $5 \times 10^5$ cells ml⁻¹ in RPMI medium supplemented with 15% FBS (FND500, ExCell Bio), 31.25 g ml⁻¹ LPS (L2630, Sigma) and 25 ng ml⁻¹ of IL4 (CK15; Novoprotein), or 31.25 g ml⁻¹ LPS alone. Cells were collected every day up to day 4 and were stained with surface markers to access the

antibody class. The flow cytometric data were analyzed with FlowJo X 10.0.7R2. The gate strategies are showed in Supplementary Fig. 14.

**Gene deletion in CH12F3 cell lines**. For gene deletion, a pair of single guide RNAs (sgRNAs), which flank one or two exons, were designed with SSC program[63]. A green fluorescent protein (GFP)-expressing plasmid and pX330-based CRISPR/Cas9 plasmids were co-transfected into CH12F3 cells. At 24 h after transfection, GFP-high cells were sorted with BD FCAS Aria II and plated into single clones in 96-well plates. Individual clones were genotyped by PCR and positive clones were further confirmed by western blot or RT-qPCR. Western blotting images were processed with ImageJ 1.52a.

**CH12F3 cell line CSR assay and drug sensitivity assay**. CH12F3 cell lines were stimulated with anti-CD40 (16-0402-86; Ebioscience; 1 µg ml⁻¹), TGF-β (CA59; Novoprotein; 0.5 ng ml⁻¹), and IL4 (CK15; Novoprotein; 5 ng ml⁻¹) for the indicated times[64]. For testing drug sensitivity[65], cells were plated at a concentration of $6 \times 10^4$ cells ml⁻¹ with indicated chemicals or different doses of treatments and viability was tested with a Cell Counting Kit-8 assay (K1018; APExBio). Cell cycle was monitored with an EdU cell proliferation detection kit (C10338-3; Ribobio). Cell apoptosis was monitored with a cell apoptosis assay kit (V13245; Invitrogen).

**HTGTS and data analysis**. HTGTS was performed according to a published protocol[66]. In brief, genomic DNA was extracted from cells. Ten micrograms of DNA was fragmented via sonication by using a Bioruptor UCD-300 and the fragmented DNA (100–2000 bp) was used as templates for linear PCR amplification with a biotin primer (Supplementary Table 1). Single-stranded PCR products were purified using Dynabeads MyOne C1 streptavidin beads (65001, Invitrogen) and ligated to the adaptor (Supplementary Table 1). Adaptor-ligated products were PCR amplified. The PCR product was further tagged with illumine P5 and P7 index primers, size-selected via gel extraction (300 ~ 800 bp) and subjected to sequencing. In primary B cells, S region rearrangements were cloned from endogenous AID-initiated Sμ breaks with 5′-RED-Iμ primer as described previously[37]. In CH12F3 cells, one IgH allele (nonproductive DJ allele) already underwent CSR (5′-DJ-Sμ/Sα-Ca-3′), whereas on the other allele (VDJ allele) IgH C region genes are in WT configuration. Thus, the Iμ primer used in CSR-activated B cells cannot be used in CH12F3 cells. Thus, a CRISPR/Cas9 break at Iγ3 region was introduced, which can help to capture AID-initiated breaks on the productive VDJ allele during CSR from IgM to IgA. Designer endonucleases, including I-Sce1 and Cas9, are the most frequently used tools to generate bait breaks in HTGTS technology[67,68]. The sgRNA sequence and the HTGTS cloning primers are listed in Supplementary Table 1. The data were analyzed with a previous published pipeline[66], in which the

raw reads were first aligned to the mouse genome (mm9) and the prey sequence was extracted. Translocation junctions were identified with the alignment files and a BED file was exported. To plot the S region resection ratio[37], we calculated the junction numbers at both the S region (s) and a 4 kb downstream region (ds), and calculated the resection ratio as: $s/(s+ds)$.

**Mutation analysis of 5′Sμ region in CH12F3 cells and CSR-activated B cells**. The 5′-region of Sμ was PCR-amplified from gDNA of CH12F3 cells and CSR-activated B cells with indicated primers in Supplementary Table 1. The PCR product was further tagged with illumine P5 and P7 index primers and subjected to illumina HiSeq or MiSeq. Demultiplexed PE150/PE250 reads were processed with SHM pipeline as described[39]. The raw reads were first aligned to the reference amplicon sequence, and substitutions/deletions/insertions were called with the SAM file. The SHM pipeline reports the mutation frequency of each nucleotide. Inside the sequenced region, sites with a mutation frequency higher than 0.1% were picked for mutation spectrum analysis.

**Knockout efficiency assessment**. Two sets of primers were designed to access the *Rev7* knockout efficiency. One amplicon is located inside of floxed region on *Rev7* gene on chr4 (Amplicon1), whereas the control amplicon is located in *Gapdh* gene on chr6 (Amplicon2). The exact chromosome coordinates (mm10) are chr4:148142786-149142897 and chr6:125163235-125163391. WT genomic DNAs with different concentrations were used as standards to validate the assay. When testing the knockout efficiency of each sample (gDNA from $Cd19^{cre}Rev7^{fl/fl}$ B cells), a WT control (gDNA from $Rev7^{fl/fl}$ B cells) was always included. The fraction of floxed allele is calculated as: $2^\wedge - [(Ct^{Amplicon2-in-control} - Ct^{Amplicon1-in-control}) - (Ct^{Amplicon2-in-sample} - Ct^{Amplicon1-in-sample})]$. In this context, knockout efficiency negatively correlates with the fraction of floxed allele.

**GC B-cell purification and SHM assay**. $B220^+PNA^{hi}$ GC B cells were sorted with BD FCAS Aria II from Peyer's Patches or spleens of SRBC-immunized mice. $J_H4$ and Jκ5 introns were PCR amplified with the indicated primers and the ~1.2 kb $J_H4$ fragment[52] and ~0.8 kb Jκ5 fragment[53] were gel-purified. The PCR products were further tagged with illumine P5 and P7 index primers and subjected to illumina HiSeq. Data were analyzed as similar as performed with S region mutations.

**Statistical analysis**. The data in the figures are presented as the mean ± SD, unless otherwise indicated. Statistical analyses were performed using R (Version 3.5.1, R Foundation for Statistical Computing, Vienna, Austria, URL http://www.R-project. org), GraphPad Prism 7 software, or Microsoft Excel (v16.16.21). The number of replicates and statistical test procedures are indicated in the figure legends. Two-tailed paired or unpaired *t*-test, or one-way analysis of variance (ANOVA) followed by Dunnett's multiple comparison test was performed if not specific noted. A *P*-value < 0.05 was considered significant and denoted as single asterisk, and $P < 0.01$, $P < 0.001$, $P < 0.0001$ were denoted as two, three and four asterisks. Statistics including sample sizes (*n*), *P*-values, *F*-values and degrees of freedom for ANOVA, and *t*-values and degrees of freedom for *t*-tests for each panel are listed in Supplementary Table 2.

**Reporting summary**. Further information on research design is available in the Nature Research Reporting Summary linked to this article.

## Data availability

HTGTS and SHM sequencing data have been deposited in the NCBI Sequence Read Archive (SRA) with the BioProject accession code: PRJNA590097. The source data underlying Figs. 1a–c, e–g, 2a–f, 3a–g, 4a–e, 5a–f, and 6a, b, and Supplementary Figs. 1a, b, d–i, 2c, 3c, 4a–f, 5a, 6b, c, 7b, 9a, b, 11a–f, 12a–c, and 13a, b are provided as a Source Data file. All other relevant data are available in the Article, Supplementary Information, or from the corresponding author upon reasonable request. Source data are provided with this paper.

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

## Acknowledgements

We thank Drs. Frederick W. Alt, Klaus Rajewsky, Kefei Yu, and Tasuko Honjo for providing reagents/protocols, and Drs. Junjie Chen and Bin-Bing Zhou for critical reading of the manuscript. This work was supported by National Key R&D Program of China (2017YFA0506700), National Natural Science Foundation of China (31670929 and 81622022), Strategic Priority Research Program of Chinese Academy of Sciences (XDB19030000), and the Chinese National 973 Project (2013CB911003).

## Author contributions

D.Y., J.C., Y.Z., M.H., S.F., X.X., Y.C., Y.S., and T.G. performed experiments and analyzed data with inputs from L.S., X.W. Y.S. and W.X. made the *Rev7flox* mouse model. J.H., J.D., and L.S.Y. provided HTGTS and SHM experimental and computational pipelines. F.L.M. designed and supervised the project, analyzed and interpreted data, and wrote the paper. All authors edited the manuscript.

## Competing interests

The authors declare no competing interests.
