## [Peer Review File · Nature Communications]

Reviewers' comments:

Reviewer #1 (Remarks to the Author):

The manuscript by Yang et al. sets to dissect the different contributions of Rev7 to AID-induced diversification reactions, Class Switch Recombination (CSR) and Somatic Hypermutation (SHM). The authors generated a conditional mouse model of Rev7, as well as several CH12 derivative cell lines deficient for Rev7 and known Rev7 interactors and functional partners. The authors proceed to analyze CSR, resection, S region mutation load, G2/M transition, and apoptosis in these model systems as well as after reconstitution with a panel of Rev7 mutants. They conclude that Rev7 supports survival of B cells following AID-induced DNA lesions, a function that is independent of its role in DSB end protection, but dependent on the interaction of Rev7 with Rev3l. Furthermore, Yang et al. shows that Rev7-deficient mice have reduced levels of SHM, thus implicating Rev7 also in this AID-initiated process.

Overall the experiments are properly designed and executed, with adequate controls and detailed statistical analysis. Furthermore, the authors employ two model systems, CH12 cell lines and splenocytes cultures from Rev7^{fl/fl}/Cd19Cre^{+/+} mice, thus reinforcing their findings. The rescue data of the cell death defect with Aicda-deficiency is convincing, and the idea that Rev7 might have a role in SHM, as well as contributing multiple functions to CSR is an interesting one, also in view of the widespread use of CSR as a read-out for the end resection regulation function of newly-identified DSB repair factors.

However, there are several major concerns that preclude publication in the present form, highlighted below.

1. For the first half (data in Fig. 1 to 3), the paper is structured in a confusing manner, with data presented in a generally descriptive way that tends to list the multiple phenotypes exhibited by the different genotypes/mutants, rather than with a step-wise layout of sequential findings. This is also reflected in the Figure titles, which lack an assertive format, thus rendering difficult to extrapolate the key finding and message of each figure. The manuscript would greatly benefit from grouping data in a manner that highlights the specific point conveyed by each figure, e.g. "Rev7 is required to promote survival of B cells during CSR", etc. The second half of the paper (findings in Figure 4 to 7) is more linear and easier to follow.

2. The authors conclude that Rev7-Rev3l interaction is required for protection of B cells against apoptosis by combining the gene knock-out results with the dissection of critical residues in Rev7, and employ Y63A-W171A double mutant in their analysis. However, residue Y63 is required for Rev7 interaction with both Shd3 and Rev3l, whereas W171A mutation disrupts the interaction of Rev7 with Rev3l but leaves the interaction with Shd3 unaffected (Ghezraoui et al., Nature 2018). Why have the authors not analysed W171A single mutant in their battery of assays (especially apoptosis & cell number)? The authors actually show reconstitution of primary B cells from Rev7^{fl/fl}/Cd19Cre^{+/+} with W171A single mutant (Fig. 3a) but they don't use it in the analyses, which is surprising since W171A would be the perfect separation of function mutant between the 53BP1-Rif1-Shieldin-mediated DNA end protection and Rev3l-dependent activities of Rev7. In this regard, it is interesting to note that in Ghezraoui et al. (Nature 2018), W171A has a partial defect in CSR, which resembles the defect shown by Rev3l-deficient CH12 cells (Fig. 2b).

3. The authors show that Rev7-deficient B cells have lower levels of SHM. How much is the underlying mechanism related to the lower mutation rate exhibited by Rev7^{fl/fl}/Cd19Cre^{+/+} at 5' S_μ? The fact that deletion of Rev7 in Ung- and Msh2-double deficient CH12 cells (Fig. S4) does not alter the mutation frequency in this region compared to control counterpart indicates that Rev7 is not required for AID targeting. However, lower mutation rates at 5' S_μ might be due to deletion of hyper-mutated/targeted sequences because of the resection phenotype, which the authors take into consideration, but also a defect in the conversion of AID-initiated lesions into DSBs, which

would be in agreement with a role of Rev7 in converting AID lesions during SHM as well. What is the frequency of AID-induced DSBs in Rev7-deficient cells? The resection phenotype precludes the assessment of DSBs formation at S regions in Rev7-deficient cells, however, this problem could be circumvented by comparing DSBs formation (and mutation frequency) in Rev7-deficient B cells reconstituted with WT versus W171A single mutant.

Minor:

1. The colors and symbols chosen for different genotypes in figure 2 are very similar, and the data is difficult to read.
2. There are some abbreviations in the text that are either specified multiple times (e.g. germinal centers (GC) on page 3, lines 51 and 68), or used before the spelled-out form (e.g. page 3, lines 59 and 60, "... at upstream S μ region and downstream Switch (S) regions.").

Reviewer #2 (Remarks to the Author):

In the manuscript entitled "Rev7 is required for processing AID-initiated DNA lesions in activated B cells", Yang et al examine class switching and somatic hypermutation in Rev7-deleted B cells. While Rev7 has been studied over the past many years as a translesion polymerase, it has acquired renewed attention as a factor in the 53BP1/Rif-1/shieldin DNA repair pathways. Loss of Rev7 in B cells has also been shown by others to impair CSR. However, the requirement of Rev7 in SHM and its mechanistic role in promoting CSR have not been fully elucidated. In this manuscript, the authors delete Rev7 in B cells and demonstrate that its loss impairs B cell survival, and this effect could be rescued by deleting AID. They show that Rev7 is required for translesion synthesis across AP sites generated following activities of AID/APOBEC and UNG. The authors conclude that Rev7 is required for end-processing during CSR as well as for B cell survival. Overall, given the interest in translesion polymerases and end-processing/end-protection complexes, this study will be of much interest across a wide range of scientists. However, the authors need to address issues described below to raise the quality of their study.

Fig. 1. The authors show that Rev7 deficient cells are arrested in the G2/M phase compared to wt cells, yet there is no defect in cell proliferation. How do they explain that? There is no quantification of the CFSE dilution plot (e.g., % and number of live cells in each bin/division) hence making their claim that no proliferation defect was seen to be unconvincing. Also, can the arrest in the G2/M phase contribute to the higher rate of apoptosis in Rev7 deficient cells? Can the increased rate of apoptosis they observe in Rev7^{-/-} fully account for the decrease in live Rev7^{-/-} cells? The authors should provide a better quantification of the rate of proliferation in Rev7 deficient vs. control cells since the data in Sup Fig 1h is very unclear. Additionally, the immunoblot in Figure 1f is really overexposed (or there is too much protein), making it difficult to conclude if there are changes in AID expression.

Fig. 2. The authors argue that Shieldin/Rif1/53bp1 and Rev7 act in the same pathway for CSR; however, they need more data to back up this claim. For example, is Rev7 recruited to the switch regions and is the recruitment dependent on Shieldin, Rif1 or 53bp? Additionally, the way the authors present their statistical significances in this figure (and throughout the manuscript) is confusing. The authors did not state clearly in the figure legend the control group that is being compared to in their statistical analysis—for example, in figures 2b and 2c, based on the asterisks, the genotype that is compared to is the WT CH12F3 cells but in figure 2d, Rev7-KO cells becomes the reference group, as shown how the trend of the statistical significance has inverted vs. figures 2b and 2c. Figure legend doesn't say why such a change is made. Even though this doesn't change the conclusion of the finding, it seems peculiar that the statistical comparisons are different.

Fig. 3. The authors observe different protein levels from their overexpression experiments (blot in Fig 3a), making it difficult to confidently compare the effects of the different mutations. Is there any way the authors could express the different mutants at roughly similar levels?

Fig 4. In the graphs with the y-axis titled 'Fraction of floxed allele (%)' [Figures 4d and 5d], it's not

mentioned at all in the Methods section or in the figure legend how the quantification of this genotyping experiment was made. What is the denominator here? The intensity of the floxed allele band relative to the deleted band? The fraction of sorted single-cells containing the deleted allele band vs. the floxed allele band? I assume from the legend that they took the bulk population and did PCR on them, so the second possibility is unlikely. In the schematic Fig 4a, the primers used for the genotyping of the floxed allele land between the loxp sites, and so, when the loxp sites recombine, no amplification would be seen. But, there's no mention of the primers that detect the deleted allele so I can't understand how are they calculating the fraction of the floxed allele in the population of cells if in their PCR they would expect either an intense band, a faint one, or no bands at all. It is also surprising that they did not have littermate controls for their CD19-Cre Rev7-fl/fl AID-KO mice, i.e., CD19-Cre Rev7-fl/fl AID-Het or AID-WT mice. The genotype that they compare to claim that AID deficiency rescues the cell death phenotype in their Rev7-deficient is the AID-KO only mice, in that the percentage of cell death between AID-KO and AID-/REV7-Double Knockout mice are similar.

Fig. 5. The authors have 6/7 replicates for panels a and b but they have only 3 replicates for e and f – maybe this could explain why they see a statistical significance in a and b but not in e and f? The authors claim that Rev3 and Rev7 act together in repairing the AID induced damage. They should probably replicate some of the findings from Figure 5 in Rev3-/- mice to support that.

Fig. 6. If the authors claim that Rev3 and Rev7 act together in filling in the abasic sites created by Ung, then they should be able to replicate these results in Rev3-/- mice.

Fig. 7. The authors use the term 'activated B cells' and CH12F3 cells interchangeably in their ex vivo analysis so it is unclear whether they are using naïve splenic B cells or CH12F3 cells in their assays (applies to other parts of the paper too)? Additionally, there also seems to be an overall increase in apoptosis within the wt and the 53bp1-/- cells from EV to AID to A3A. Could they make statistical tests between the grey columns only or the blue columns only? (It just seems like they were cherry picking by only comparing the red columns).

Reviewer #3 (Remarks to the Author):

The manuscript submitted from Yang and colleagues addresses the role of Rev7 in programmed DNA breaks in mature B cells. This group has examined the outcomes of AID-mediated genetic modifications in B cells undergoing CSR and/or SHM both in a mouse model with lineage specific inactivation of Rev7, and in cells lines engineered with mutations in Rev7 and associated repair partners. They, as several other groups have previously reported, find that Rev 7 is necessary for CSR and in the control of end-resection of AID-mediated S region breaks. They then seek to understand the role of Rev7 in SHM and focus on mutations of the 5' S_μ region in activated B cell using an established high-throughput sequencing approach. They find that the mutational patterns are altered (C>G transversion was significantly decreased in Rev7 deficiency) pointing to a Rev7 role in the process. They also found increased cell death and G2-M arrest in activated Rev7 B cells which was largely AID-dependent. Using engineered cell lines they then implicate a Rev1-Rev7 interaction in mediating SHM.

The role of Rev 7 in the genetic modifications of B cells is of high interest, and this system is valuable for elucidating the physiological roles of Rev7 in general. The tools and approaches presented are innovative and help further elucidate Rev7 function. However, as presented, the data do not unequivocally support the conclusions of the study. SHM and CSR are tightly linked processes and it is difficult to conclude what (lesion repair outcome in Rev-7 deficiency) is SHM specific and what is related to the CSR defects seen in Rev7 deficiency.

Specific comments:

1- The authors largely rely on the 5' S μ region for SHM analysis. While of some interest, this is not the usual region assayed for SHM which is typically performed in V regions. The IgH J and IgK regions did not have changes in the mutation spectrum in Rev7-deficient GC B cells. This latter finding should be further analyzed and carefully dissected as this is critical to understanding SHM in Rev7 deficiency

2- It is apparent that there is a large selective pressure for Rev7 proficiency in activated B cells, likely due to the CD19 driven Cre. This is a critical point and markers for deletion of Rev7 should be considered to be able to definitively link mutation outcome to Rev7 deficiency.

3- Most of the mutation data is generated from loci which are tightly linked to the negative outcomes of Rev7 deficiency after CSR (i.e. 5' S μ where long resections can delete this region). One potential way to assay for SHM in B cells that haven't undergone CSR is to sort for IgM expressing B cells and look for V region mutation load and the associated pattern. Sequencing both productive and non productive rearrangements linked with unswitched alleles (e.g. RNA seq) could also help generate useful data to support SHM-specific mutation outcomes.

4- The G2/M phase arrest should be better described in CH12 cells as this does not appear to be AID-dependent.

5- Cell death is linked to AID-deamination which leads to CSR and SHM (in vivo). The relative contribution of each process is unclear when both are activated as is the case in GC's. This is not clear and should be dissected. Looking at unswitched B cells for SHM and having markers for Rev7 deletion would help this analysis.

6- The use of CRISPR/Cas9-generated breaks at the Ig3 region of CH12 cells is unclear and should be better justified.

7- The base editing data are of some interest but are largely undeveloped and speculative, and outside the focus of the study.

#NCOMMS-19-23548A

Response to Reviewer's comments (original reviewers' comments are in italic):

Reviewers' comments:

Reviewer #1 (Remarks to the Author):

The manuscript by Yang et al. sets to dissect the different contributions of Rev7 to AID-induced diversification reactions, Class Switch Recombination (CSR) and Somatic Hypermutation (SHM). The authors generated a conditional mouse model of Rev7, as well as several CH12 derivative cell lines deficient for Rev7 and known Rev7 interactors and functional partners. The authors proceed to analyze CSR, resection, S region mutation load, G2/M transition, and apoptosis in these model systems as well as after reconstitution with a panel of Rev7 mutants. They conclude that Rev7 supports survival of B cells following AID-induced DNA lesions, a function that is independent of its role in DSB end protection, but dependent on the interaction of Rev7 with Rev3l. Furthermore, Yang et al. shows that Rev7-deficient mice have reduced levels of SHM, thus implicating Rev7 also in this AID-initiated process.

Overall the experiments are properly designed and executed, with adequate controls and detailed statistical analysis. Furthermore, the authors employ two model systems, CH12 cell lines and splenocytes cultures from Rev7^{fl/fl}Cd19Cre⁺ mice, thus reinforcing their findings. The rescue data of the cell death defect with Aicda-deficiency is convincing, and the idea that Rev7 might have a role in SHM, as well as contributing multiple functions to CSR is an interesting one, also in view of the widespread use of CSR as a read-out for the end resection regulation function of newly-identified DSB repair factors.

Response: We thank the reviewer for these very positive comments and constructive suggestions. In the following paragraphs, we address all the concerns point-by-point with new experimental data and further editing of the manuscript. We hope the reviewers will find the revised manuscript much improved.

However, there are several major concerns that preclude publication in the present form, highlighted below.

1. For the first half (data in Fig. 1 to 3), the paper is structured in a confusing manner, with data presented in a generally descriptive way that tends to list the multiple phenotypes exhibited by the different genotypes/mutants, rather than with a step-wise layout of sequential findings. This is also reflected in the Figure titles, which lack an assertive format, thus rendering difficult to extrapolate the key finding and message of each figure. The manuscript would greatly benefit from grouping data in a manner that highlights the specific point conveyed by each figure, e.g. "Rev7 is required to promote survival of B cells during CSR", etc. The second half of the paper (findings in Figure 4 to 7) is more linear and easier to follow.

Response 1: We fully agree with this comment and reorganized the first half in the revised manuscript.

In brief, we introduce our finding that "REV7 deficiency leads to B cell death during CSR" at the beginning. Next, we use gene knockout cell lines to dissect the potential pathways contributing to this function, and report "REV7 and REV3L protect activated CH12 cells from cell death". Later, we use a panel of REV7 mutants including separation-of-function mutants as this reviewer suggested to further address the critical role of REV7 HORMA domain in promoting activated-B cell survival. To make the manuscript more

readable, we changed panel orders, text and Figure titles accordingly. We hope the reviewers will find the revised version much easier to follow.

2. The authors conclude that Rev7-Rev3l interaction is required for protection of B cells against apoptosis by combining the gene knock-out results with the dissection of critical residues in Rev7, and employ Y63A-W171A double mutant in their analysis. However, residue Y63 is required for Rev7 interaction with both Shld3 and Rev3l, whereas W171A mutation disrupts the interaction of Rev7 with Rev3l but leaves the interaction with Shld3 unaffected (Ghezraoui et al., Nature 2018). Why have the authors not analysed W171A single mutant in their battery of assays (especially apoptosis & cell number)? The authors actually show reconstitution of primary B cells from Rev7^{fl/fl}Cd19^{Cre/+} with W171A single mutant (Fig. 3a) but they don't use it in the analyses, which is surprising since W171A would be the perfect separation of function mutant between the 53BP1-Rif1-Shieldin-mediated DNA end protection and Rev3l-dependent activities of Rev7. In this regard, it is interesting to note that in Ghezraoui et al. (Nature 2018), W171A has a partial defect in CSR, which resembles the defect shown by Rev3l-deficient CH12 cells (Fig. 2b).

Response 2: We thank this reviewer for this constructive suggestion. We performed new experiments and now put the single-mutant data into the revised manuscript.

We confirmed the phenotypes of the separation-of-function W171A mutant in CSR (Ghezraoui et al., Nature 2018). During CSR, the W171A mutant resulted slightly decreased CSR level and non-expanded resection, while the Y63A mutant resulted much severe CSR defect and expanded resection (Fig. 3b-c). Together with the another Shieldin interaction mutant K129A, these single mutants nicely separated REV7's role in 53BP1-RIF1-Shieldin pathway from others.

As observed from protein structure and biochemical data, REV7 interacts with REV3L through both Y63 and W171, and a W171A mutant still interacts with REV3L *in vivo* (Hara et al., J Biol Chem 2010). The differential contribution of Y63 and W171 in REV7-SHLD3 and REV7-REV3L interactions are also compared in a recent structural study (Dai et al., J Biol Chem 2019). Thus, the W171A mutant mimic decreased REV7-REV3L interaction as demonstrated by the milder CSR defect (CSR level: Rev3^{fl/fl}, ~50% of WT level; REV7-W171A, ~80% of the level in WT-complemented cells). The partial loss of REV7-REV3L interaction with W171A mutant also result in a mild decreased S region mutation frequency as discussed below (Response 3 to this reviewer). However, the W171A single mutation did not significantly affect live cell numbers or apoptosis (Fig. 3f-g), suggesting the remaining interaction of REV7-REV3L is sufficient to support cell survival. In this context, W171A mutant is not a perfect/complete separation-of-function mutant in distinguishing REV7's role in REV7-REV3L pathway from others. In the revised manuscript, we added this new piece of data and also more details in the supplementary data. The new data with REV7 single mutants (together with the data mentioned in the Response 3 to this reviewer) help to clarify the different function outputs of various REV7 mutants (Fig. 3), which make our conclusion stronger.

3. The authors show that Rev7-deficient B cells have lower levels of SHM. How much is the underlying mechanism related to the lower mutation rate exhibited by Rev7^{fl/fl}Cd19^{Cre/+} at 5' Sμ? The fact that deletion of Rev7 in Ung- and Msh2-double deficient CH12 cells (Fig. S4) does not alter the mutation frequency in this region compared to control counterpart indicates that Rev7 is not

required for AID targeting. However, lower mutation rates at 5' S μ might be due to deletion of hyper-mutated/targeted sequences because of the resection phenotype, which the authors take into consideration, but also a defect in the conversion of AID-initiated lesions into DSBs, which would be in agreement with a role of Rev7 in converting AID lesions during SHM as well. What is the frequency of AID-induced DSBs in Rev7-deficient cells? The resection phenotype precludes the assessment of DSBs formation at S regions in Rev7-deficient cells, however, this problem could be circumvented by comparing DSBs formation (and mutation frequency) in Rev7-deficient B cells reconstituted with WT versus W171A single mutant.

Response 3: We agree with this reviewer that SHM in REV7 deficiency and the potential role of REV7 in conversion of AID-initiated lesions into DSBs need further describing and exploring. In this context, we have performed new experiments and further discussed our data to clarify REV7's functions.

- 3a. As both this reviewer and the Reviewer 3 pointed out, the mutation of 5'S μ in CSR-activated B cells is not an equivalent assay for SHM at *Ig V* regions in germinal center B cells. There are many differences between the two events, including the intrinsic sequence, downstream DNA repairs, etc. However, mutation spectrum analysis of 5'S μ during CSR could be a simple assay to study C>G transversion in B cells, as shown in the REV deficient B cells (Fig. 2d). In the revised manuscript, we clarified the usage of this assay (Page 7, first paragraph) and also kept the discussion on mutation frequency of 5'S μ during CSR minimal. We also further analyzed the SHM at *Ig V* regions including J κ 5 intron which locus is not affected by CSR in REV7 deficient GC B (Supplementary Fig. 11). We describe these results and discuss the implications in Page 11-12 of the revised text.
- 3b. We agree with this reviewer that the lower mutation frequency at 5'S μ during CSR could be contributed both by expanded resection and/or by REV7's role in converting AID lesions. As suggested by this reviewer and revealed by the updated data, *Rev7* KO B cells complemented with W171A mutant showed decreased 5'S region mutation frequency and CSR levels, comparing to WT-complemented B cells (Supplementary Fig. 9a). As discussed above, REV7 W171A mutant is a perfect mutant to separated REV7's role in 53BP1-RIF1-Shieldin pathway from others. We also compared the DSB levels between *Rev7* KO B cells complemented with WT and W171A by using HTGTS assay. It is of note that HTGTS assay does not directly report the amount of DSBs but reflect the number through gene rearrangement (Chiarie *et al.*, Cell 2011). The nearly wild type level end resection in *Rev7* KO B cells complemented with W171A mutant (Fig. 3c) allows the comparison. We found the S μ -S γ 1 junction number significantly decreased in the W171A cells (Supplementary Fig. 8b, S μ -S γ 1 junctions decreased from 51% to 39% of total S μ junctions). Thus, we conclude that REV7's roles in both inhibiting end resection and converting AID-lesions contribute to the decreased mutation level at 5'S μ region.

We really appreciate the suggestion to explore the different possibility and use REV7 mutant to answer the related questions, which, we feel, greatly improve the quality of our study.

Minor:

1. The colors and symbols chosen for different genotypes in figure 2 are very similar, and the data is difficult to read.

Response 4: We are sorry for the unclear presentation. We redraw the panels with more colors and bigger symbols. In the panels, we used similar colors to indicate genes in the same pathway, e.g. *53bp1*^{-/-}, *Rif1*^{-/-} and *Shld3*^{-/-} are illustrated in dark blue, blue and light blue. For Panel A, several lines are overlapped in the survival curves since these genotypes display similar sensitive to the treatment, which can be clearly visualized in the AUC plots. We also do our best to uniform the colors among figures, e.g. all *Rev7* KO cells are illustrated in red in Fig. 1, 2, 4 and 5.

2. There are some abbreviations in the text that are either specified multiple times (e.g. germinal centers (GC) on page 3, lines 51 and 68), or used before the spelled-out form (e.g. page 3, lines 59 and 60, "... at upstream *Sμ* region and downstream Switch (S) regions.").

Response 5: We thank this reviewer for the careful reading of our manuscript. We have corrected all the similar errors. In the revision, we also corrected the gene and protein symbols to follow the mouse gene nomenclature guidelines.

Reviewer #2 (Remarks to the Author):

In the manuscript entitled "Rev7 is required for processing AID-initiated DNA lesions in activated B cells", Yang et al examine class switching and somatic hypermutation in Rev7-deleted B cells. While Rev7 has been studied over the past many years as a translesion polymerase, it has acquired renewed attention as a factor in the 53BP1/Rif-1/shieldin DNA repair pathways. Loss of Rev7 in B cells has also been shown by others to impair CSR. However, the requirement of Rev7 in SHM and its mechanistic role in promoting CSR have not been fully elucidated. In this manuscript, the authors delete Rev7 in B cells and demonstrate that its loss impairs B cell survival, and this effect could be rescued by deleting AID. They show that Rev7 is required for translesion synthesis across AP sites generated following activities of AID/APOBEC and UNG. The authors conclude that Rev7 is required for end-processing during CSR as well as for B cell survival. Overall, given the interest in translesion polymerases and end-processing/end-protection complexes, this study will be of much interest across a wide range of scientists. However, the authors need to address issues described below to raise the quality of their study.

Response: We thank this reviewer for the appreciation of the significant and the encouraging comments. In the revision, we performed new experiments and present more data to address this reviewer's concerns.

Fig. 1. The authors show that Rev7 deficient cells are arrested in the G2/M phase compared to wt cells, yet there is no defect in cell proliferation. How do they explain that? There is no quantification of the CFSE dilution plot (e.g., % and number of live cells in each bin/division) hence making their claim that no proliferation defect was seen to be unconvincing. Also, can the arrest in the G2/M phase contribute to the higher rate of apoptosis in Rev7 deficient cells? Can the increased rate of apoptosis they observe in Rev7^{-/-} fully account for the decrease in live Rev7^{-/-} cells? The authors should provide a better quantification of the rate of proliferation in Rev7 deficient vs. control cells since the

data in Sup Fig 1h is very unclear. Additionally, the immunoblot in Figure 1f is really overexposed (or there is too much protein), making it difficult to conclude if there are changes in AID expression.

Response 1: We thank this reviewer for the critical comments. We now have updated data to support our conclusions.

1a: We showed that REV7 deficient activated B cells had a significant growth defect at Day 4 by cell number counting (Fig. 1b) or flow cytometry (Supplementary Fig. 1i). The conclusion is different from the recently report by using a fluorescent dye-based cell proliferation assay (Ghezraoui et al., Nature 2018). In our study, we also performed a similar CFSE-integration based proliferation assay. In the revision, we carefully analyzed CFSE dilution plots and made more quantification analysis according to this reviewer's suggestion (Supplementary Fig. 1i). We compared cell division by two-tail paired *t* test of three biological replicates, as in each time B cells from a pair of mice are assayed.

Although no severe division defect was observed in REV7 deficiency, a slightly but significant decreased "Division 6" population were observed in REV7 deficient CSR-activated B cells (Supplementary Fig. 1i, middle). The discrepancy of CFSE-based assays with cell counting may come from the different experimental procedures. In CFSE assay, only "live" cells are subjected to CFSE dye analysis, while the information of the significant decreased "live" population in the forward and side scatter plot failed to be displayed in the CFSE assay. The slight defect of cell division could be contributed by the apoptosis or G2/M arrest. We reorganized the result section and briefly discussed the issue in the discussion section. We thank this reviewer for this specific suggestion to make our conclusion much clear.

1b: The G2/M phase arrest is not a major contributor of the higher apoptosis rate in REV7 deficient cells. As we showed in the Fig. 4e, *Aicda* deletion can fully rescue the apoptosis phenotype but not G2/M arrest. Thus, the apoptosis mainly attributes to the unrepaired AID-dependent DNA lesions. We made the point clear in the revision (Page 11, first paragraph).

1d: To better display the AID protein levels, we now present new immunoblot data of AID expression with serial diluted samples to make it more quantitative (Supplementary Fig. 1f). We conclude that AID protein level is not changed by deleting *Rev7*.

Fig. 2. The authors argue that Shieldin/Rif1/53bp1 and Rev7 act in the same pathway for CSR; however, they need more data to back up this claim. For example, is Rev7 recruited to the switch regions and is the recruitment dependent on Shieldin, Rif1 or 53bp? Additionally, the way the authors present their statistical significances in this figure (and throughout the manuscript) is confusing. The authors did not state clearly in the figure legend the control group that is being compared to in their statistical analysis—for example, in figures 2b and 2c, based on the asterisks, the genotype that is compared to is the WT CHI2F3 cells but in figure 2d, Rev7-KO cells becomes the reference group, as shown how the trend of the statistical significance has inverted vs. figures 2b and 2c. Figure legend doesn't say why such a change is made. Even though this doesn't change the conclusion of the finding, it seems peculiar that the statistical comparisons are different.

Response 2: We thank this reviewer for the critical comments, and also appreciate the reviewer's comment that the unclear statistics "doesn't change the conclusion of the finding".

Here, we performed experiments to address this reviewer's question and also update the data with proper statistics.

2a: REV7 is the central subunit of Shieldin complex which also contains SHLD1/2/3. The Shieldin complex functions as an effector of 53BP1-RIF1 pathway. In CSR, we showed REV7 functions in the same pathway with 53BP1/RIF1/Shieldin as REV7-53BP1/RIF1/Shieldin double deficiencies result to similar CSR levels as those in 53BP1/RIF1/Shieldin single deficiencies (Fig. 2b). The end resection in S regions also suggested the same conclusion (Fig. 2c). However, we also carefully dissected the multiple roles of REV7 in CSR, and found its functions in TLS also help processing S region lesions, as suggested by the altered mutation spectrums in REV7-53BP1/RIF1/Shieldin double deficiencies (Fig. 2d). Thus, REV7 could be recruited to S regions through both DSB response and TLS pathways, and we speculate that REV7 could be recruited to S regions in both Shieldin-dependent and -independent manner.

We agree with this reviewer that a direct demonstration of S region recruitment could be an add-on. Thus, we performed ChIP experiment in WT and KO B cells with a commercial anti-REV7 antibody (#A9861; Abclonal). Although the antibody is successfully used in western blot experiments in our study (Fig. 3a, Supplementary Fig. 1a and Supplementary Fig. 5a), the chromatin immunoprecipitation with this antibody did not work. Since we routinely performed ChIP-qPCR with an anti-AID antibody and no REV7 ChIP data have been reported so far, we speculate the anti-REV7 antibody we tested cannot chromatin-immunoprecipitated the REV7 associated chromatin.

As suggested by Reviewer 1, we address this question with a panel of REV7 separation-of-function mutants. Although REV7 interacts with its many cofactors through a similar safe-belt model, the individual residues contribute differently as revealed by the structural studies (Hara et al., J Biol Chem 2010; Dai et al., J Biol Chem 2019). In this context, decreased CSR and increased end resection were observed in mutants losing interaction of other Shieldin subunits, including Y63A (abolished interaction with SHLD3) and K129A (abolished interaction with SHLD2), but not in mutant that does not affect Shieldin complex (W171A mutant resulted slightly decreased CSR level and non-expanded resection) (Fig. 3b,c). These data support that REV7 functions in the 53BP1-RIF1-Shieldin pathway to protect S region broken ends from resection.

2b: We are sorry for the unclear statement of statistic methods. In the revision, we present all the figures with clear legends and proper statistical significances. The raw data are also uploaded as a source data file. We hope the reviewer now will see an improvement.

Fig. 3. The authors observe different protein levels from their overexpression experiments (blot in Fig 3a), making it difficult to confidently compare the effects of the different mutations. Is there any way the authors could express the different mutants at roughly similar levels?

Response 3: We really appreciate the kind suggestion. We have performed additional experiments with those mutants and present new data in the revision.

First, we have independently performed the overexpression experiments multiple times under the condition where viral titer, cell numbers and culture conditions are stringently controlled, and the mutant protein levels always showed the same trend. Next,

as showed in the Fig. 3a, all the mutants were over-expressed comparing to the endogenous REV7 protein level. In this context, REV7-overexpression complement the CSR defect in *Rev7* KO CSR-activated B cells to a level comparable to that in wildtype cells (Fig. 3b), suggesting the different protein level is not the major determinate for function outcomes of REV7 mutants. Last, among the mutants, we did notice that some mutants like K129A with similar protein level to WT but showed dramatic CSR defect. Thus, we conclude that under the current experiment condition the protein level variation does not affect function assessment.

We present new western blot result obtained from experiments in which the mutants expressed at roughly similar levels to our best (revised Fig. 3a).

*Fig 4. In the graphs with the y-axis titled 'Fraction of floxed allele (%)' [Figures 4d and 5d], it's not mentioned at all in the Methods section or in the figure legend how the quantification of this genotyping experiment was made. What is the denominator here? The intensity of the floxed allele band relative to the deleted band? The fraction of sorted single-cells containing the deleted allele band vs. the floxed allele band? I assume from the legend that they took the bulk population and did PCR on them, so the second possibility is unlikely. In the schematic Fig 4a, the primers used for the genotyping of the floxed allele land between the loxp sites, and so, when the loxp sites recombine, no amplification would be seen. But, there's no mention of the primers that detect the deleted allele so I can't understand how are they calculating the fraction of the floxed allele in the population of cells if in their PCR they would expect either an intense band, a faint one, or no bands at all. It is also surprising that they did not have littermate controls for their CD19-Cre *Rev7*-fl/fl AID-KO mice, i.e., CD19-Cre *Rev7*-fl/fl AID-Het or AID-WT mice. The genotype that they compare to claim that AID deficiency rescues the cell death phenotype in their *Rev7*-deficient is the AID-KO only mice, in that the percentage of cell death between AID-KO and AID-/REV7-Double Knockout mice are similar.*

Response 4: We are sorry for the unclear illustration of our experimental design and incomplete controls. In the revision, we either present more clear method details or present new data to address these concerns.

4a. To assay the *Rev7* knock-out efficiency, we designed two sets of primers (Fig. 4a). One amplicon is located in the region flanked by LoxP (floxed region) on *Rev7* gene on chr4 (Amplicon1), while the control amplicon is located in *Gapdh* gene on chr6, (Amplicon2). The exact chromosome coordinates (mm10) are chr4:148142786-149142897 and chr6:125163235-125163391. For both amplicons, the PCR efficiencies were determined by quantitative PCR. We used wild type genomic DNAs (gDNAs) with different concentrations as standards to validate the assay, which showed one to one ratio of those two amplicons. When testing the knock-out efficiency of each *sample* (gDNA from *Cd19creRev7^{fl/fl}* B cells), we always included a wild type *control* (gDNA from *Rev7^{fl/fl}* B cells). The "floxed allele" indicates the intact *Rev7* allele with normal *Rev7* expression. The "fraction of floxed allele" is calculated as:

$$2^{-[(Ct^{Amp2/control} - Ct^{Amp1/control}) - (Ct^{Amp2/sample} - Ct^{Amp1/sample})]}$$

In this context, knock-out efficiency negatively correlates with the "fraction of floxed allele".

In the revision, we have put the detailed method description in the Method section (Page 17, second paragraph), and also redraw a schematic illustration in Fig. 4a. We thank this reviewer for the suggestion that makes our data much understandable.

4b. Regarding the littermate controls, we performed new sets of experiments and also collected the mouse information used in the current study.

Here is brief summary of the mouse breeding procedure (Rebuttal Fig. 1a). Originally, an *Aicda*^{-/-} mouse were crossed with a *Cd19*^{cre/cre}*Rev7*^{fl/fl} mouse (both mice are C57BL/6J background) to yield F1 *Aicda*^{+/-}*Cd19*^{cre/+}*Rev7*^{fl/+} mice. Ideally, we should get the correct genotypes from the intercross of F1 *Aicda*^{+/-}*Cd19*^{cre/+}*Rev7*^{fl/+} mice. However, the chance of getting enough numbers of F2 mice with proper genotypes (*Aicda*^{-/-}*Cd19*^{cre/+}*Rev7*^{fl/fl}) is low. Thus, we setup breeding pairs of F2 mice with the genotype of *Aicda*^{-/-}*Cd19*^{cre/+}*Rev7*^{fl/+} to generate the F3 mice (*Aicda*^{-/-}*Cd19*^{cre/+}*Rev7*^{fl/fl}) used in Fig. 4 and 5. In this context, all the mice were kept in the same room of the same facility, and age/gender-matched mice were used in our study.

Rebuttal Fig. 1 Breeding history of *Aicda*^{-/-}*Cd19*^{cre/+}*Rev7*^{fl/fl} mice.

a Breeding strategy of *Aicda*^{-/-}*Cd19*^{cre/+}*Rev7*^{fl/fl} mice.

b *Aicda*^{-/-}*Cd19*^{cre/+}*Rev7*^{fl/fl} mice used in our current study.

During the revision, we also performed a new set of experiments which are now showed in Fig. 4b,c. The four genotypes, including *Cd19*^{cre}, *Cd19*^{cre}*Rev7*^{fl/fl}, *Aicda*^{-/-} and *Aicda*^{-/-}*Cd19*^{cre/+}*Rev7*^{fl/fl}, were processed under same conditions and the experiment was independently replicated for three times. Same conclusion is drawn from the new set of experiments. We are confident about the observed cell death rescue phenotype revealed by these mouse models. The information of *Aicda*^{-/-}*Cd19*^{cre/+}*Rev7*^{fl/fl} mice are listed in Rebuttal Fig. 1b.

Fig. 5. The authors have 6/7 replicates for panels a and b but they have only 3 replicates for e and f – maybe this could explain why they see a statistical significance in a and b but not in e and f? The authors claim that Rev3 and Rev7 act together in repairing the AID induced damage. They should probably replicate some of the findings from Figure 5 in Rev3^{-/-} mice to support that.

Response 5: We appreciate these critical comments from this reviewer. We either present more data or have more discussion in the revision to address these concerns.

5a. For the number of mice used for germinal center B cell analysis, we now assay more mice during the revision time to support our conclusion, which are showed in Fig. 5e-f. Same conclusion is drawn from the data.

5b. Regarding the *Rev3^{fl/fl}* mice, we currently do not have the line in our facility, and the importing and breeding may take months. Thus, the proposed experiments are not feasible in a short time. However, we have characterized other genes (*Rev7*, *53bp1*) for their functions during CSR with gene-deletion CH12F3 lines or mouse models, and these two systems always yield same conclusions. Thus, we speculate that the conclusion drawn from *Rev3^{l-/-}* cell lines should be confirmed by the primary *Rev3^l* knockout B cells. Since we do not have the data from the *Rev3^{fl/fl}* mice, we carefully checked all the statements and make sure we did not overstate.

Fig. 6. If the authors claim that Rev3 and Rev7 act together in filling in the abasic sites created by Ung, then they should be able to replicate these results in Rev3^{-/-} mice.

Response 6: We fully agree that results from *Rev3^l* knockout primary B cells should strengthen our conclusions. However, the mice work will take months to complete as we do not have the mice in house. In this context, we think the data from *Rev7* knockout cell lines fully support the dissection of cell death associated with *Rev7* knockout in activated B cells. As previously reported by Tomida et al. (Nucleic Acids Res, 2015), REV7 and REV3L have independent functions inside of Polζ complex upon DNA damages. Thus, we carefully check all the statements and keep our statements precise.

Fig. 7. The authors use the term 'activated B cells' and CH12F3 cells interchangeably in their ex vivo analysis so it is unclear whether they are using naïve splenic B cells or CH12F3 cells in their assays (applies to other parts of the paper too)? Additionally, there also seems to be an overall increase in apoptosis within the wt and the 53bp1^{-/-} cells from EV to AID to A3A. Could they make statistical tests between the grey columns only or the blue columns only? (It just seems like they were cherry picking by only comparing the red columns).

Response 7: We thank this reviewer for the helpful comments.

- 7a. In the revision, we corrected all the terms to avoid any misunderstanding. Specifically, we call the naïve splenic B cells stimulated by LPS plus IL4 as “CSR-activated B cells” as previously report (Meng et al., Cell 2014). For CH12F3 cells, we use the line name, e.g. CH12F3 cells, CIT-stimulated CH12F3 cells, etc.
- 7b. As suggested by this reviewer, we did the statistical test among different conditions in the same genotype. Overexpression of AID was toxic to *Rev7^{-/-}* CH12F3 cells, while overexpression of A3A resulted significant increased apoptotic populations in all CH12F3 cell lines. However, both AID and A3A overexpression resulted in the most severe cell death in *Rev7^{-/-}* CH12F3 cells (comparing CH12F3, *53bp1^{-/-}* and *Rev7^{-/-}* cells under same treatment condition). The geno-toxicity of A3A overexpression was well-documented in many reports (Landry et al., EMBO Rep 2011; Suspene et al., PNAS 2011; Burns et al., Nature 2013), as A3A can generate spontaneous DNA deamination lesions. Thus, the basal spontaneous deamination, AID-overexpression and A3A-overexpression indicated three different levels of DNA deamination in cells. In this context, REV7 is required for resolving the unrepaired AP sites from various sources including those potentially from cytidine base editors.

As suggested by Reviewer 3, the base editing data are “*outside the focus of the study*”. We then moved the text into Discussion section and listed the panels as

Supplementary Figures, with carefully statement of the implications and proper statistical tests.

Reviewer #3 (Remarks to the Author):

The manuscript submitted from Yang and colleagues addresses the role of Rev7 in programmed DNA breaks in mature B cells. This group has examined the outcomes of AID-mediated genetic modifications in B cells undergoing CSR and/or SHM both in a mouse model with lineage specific inactivation of Rev7, and in cells lines engineered with mutations in Rev7 and associated repair partners. They, as several other groups have previously reported, find that Rev 7 is necessary for CSR and in the control of end-resection of AID-mediated S region breaks. They then seek to understand the role of Rev7 in SHM and focus on mutations of the 5' S_μ region in activated B cell using an established high-throughput sequencing approach. They find that the mutational patterns are altered (C>G transversion was significantly decreased in Rev7 deficiency) pointing to a Rev7 role in the process. They also found increased cell death and G2-M arrest in activated Rev7 B cells which was largely AID-dependent. Using engineered cell lines they then implicate a Rev1-Rev7 interaction in mediating SHM.

The role of Rev7 in the genetic modifications of B cells is of high interest, and this system is valuable for elucidating the physiological roles of Rev7 in general. The tools and approaches presented are innovative and help further elucidate Rev7 function. However, as presented, the data do not unequivocally support the conclusions of the study. SHM and CSR are tightly linked processes and it is difficult to conclude what (lesion repair outcome in Rev-7 deficiency) is SHM specific and what is related to the CSR defects seen in Rev7 deficiency.

Response: We thank this reviewer for the positive comments overall. We address the main experimental reservations raised in the revised manuscript with new data described above in response to Reviewers 1 and 2, as well as below in response to specific comments.

We believe that this reviewer's major concern partly comes from our unclear presentation. As commented by Reviewer 1, the first half of the previous manuscript described multiple roles of REV7 in antibody diversification in a descriptive way. In the revised manuscript, we have reorganized the text to emphasize on our main points. In brief, we found that REV7 is required to maintain B cell survival during CSR at the beginning (Fig.1). Next, we use gene knockout CH12F3 cell lines (Fig. 2) and a panel of REV7 mutants (Fig. 3) to dissect the potential pathways contributing to this function. Then, we found the cell death of REV7-deficient CSR-activated B cells can be rescued by *Aicda* deletion (Fig. 4). Similar phenomenon was observed in germinal center B cells *in vivo* (Fig. 5). Finally, we found UNG-processed AID-lesion is the main cause of cell death in REV7-deficiency (Fig. 6). We used mutation spectrum analysis and CSR end-joining as readouts to dissect the multiple roles of REV7, and found REV7 is required for optimal CSR and SHM. The cell death in REV7 deficiency can be contributed by both CSR and SHM, as in both processes UNG is important to process AID-lesion and its product (AP sites) is one of the major threats to genome integrity in AID-expressing B cells.

As this reviewer mentioned that CSR and SHM are tightly linked processes, we addressed the SHM-specific outcome of REV7 deficiency with different assays, e.g. checking the SHM in Jk5 intron, examining SHM in IgM⁺ population, trying to access SHM

with RNA-Seq data. We are confident that REV7 plays an important role in repairing AID-lesions during SHM. With new data and further editing of the manuscript, we hope the reviewer now will see an improvement.

Specific comments:

1- The authors largely rely on the 5' S μ region for SHM analysis. While of some interest, this is not the usual region assayed for SHM which is typically performed in V regions. The IgH J and IgK regions did not have changes in the mutation spectrum in Rev7-deficient GC B cells. This latter finding should be further analyzed and carefully dissected as this is critical to understanding SHM in Rev7 deficiency

Response 1: We agree that the mutation of 5'S μ in CSR-activated B cells is not an equivalent assay for SHM at V regions in germinal center B cells. As discussed above, these two assays have many differences. CSR and SHM are initiated by the same enzyme – AID, but yield different main outcomes (gene rearrangement for CSR, while mutation for SHM). However, mutation at S regions are also observed during CSR, and deletion happens at a relative lower frequency in SHM. The mutation spectrum analysis of 5'S μ during CSR is a simple assay to study C>G transversion in CSR-activated B cells and CH12F3 cells. In the revised manuscript, we clarified the usage of this assay (Page 7, first paragraph).

As this reviewer suggested, we focus on J_H and J_K introns in GC B cells and carefully dissect the mutation spectrum difference (Supplementary Fig. 11). The mutation rate/frequency on these regions were significantly decreased in REV7 deficiency, but no significant change in mutation spectrum was observed (Supplementary Fig. 11e-f). The different mutation spectrum changes of 5'S μ during CSR and Ig V introns during SHM in REV7 deficiency could be due to many reasons as following:

- 1). The counter-selection of REV7 deficiency in germinal center B cells. As discussed in the Response 2 to this reviewer, in the population of harvested Rev7 condition-knockout germinal center B cells, the completely Rev7 knockout B cells with high AID expression may only take a fraction.
- 2). The mutation rates in the Ig V introns are relatively lower than these in the Ig V exons as documented by many labs in the field, which could limit the finding of mild changes.
- 3). In mutation spectrum analysis of 5'S μ in CSR-activated B cells, no AT-spreading is observed, implicating a lack of Pol η function in those B cells (Zeng et al., Nat Immunol 2001). The mild changes of C>G transition in GC B cells could be masked by another error-prone polymerases e.g. Pol η in SHM. This idea is supported by the previous observation by Saribasak et al. (J Exp Med 2012) that Pol ζ 's function in SHM is masked by Pol η .

We put the new analyses in the Supplementary Figure 11, and also describe the results and discuss the implications in Page 11-12 of the revised text.

2- It is apparent that there is a large selective pressure for Rev7 proficiency in activated B cells, likely due to the CD19 driven Cre. This is a critical point and markers for deletion of Rev7 should be considered to be able to definitively link mutation outcome to Rev7 deficiency.

Response 2: We thank the reviewer for raising this question. We now incorporate new data and further discussion to address the concern.

The “*selection pressure for Rev7 proficiency in activated B cells*” correlates with the increased apoptosis in REV7 deficiency observed in either CH12F3 cells or CSR-activated B cells. In the *Cd19creRev7^{fl/fl}* B cells, the *Rev7* KO allele is counter-selected in both CSR-activated B cells and germinal center B cells. We have developed assay to monitor the knock-out efficiency as showed in Fig. 4a and updated Method section. We also cautiously interpret the mutation outcomes in CSR-activated B cells and germinal center B cells for the reasons listed in the Response 1 to this reviewer. With the *Aicda*^{-/-} rescue data, we concluded that the selection pressure come from the toxic effect of unrepaired AID-initiated lesions, as the deletion of *Aicda* not only rescue the apoptosis but also the counter-selection of REV7 deficient B cells.

We agree that markers for deletion can be done by elegant design of the knockout construct. However, the issue can be circumvented by using gene knockout CH12F3 cell lines, in which multiple independent single-cell clones were retrieved and analyzed for the outcomes. CRISPR/Cas9-derived gene deletion B cell lines have been proven to be reliable models to reveal end-joining mechanisms to avoid complicated mouse models (Kumar et al., PNAS 2016; Panchakshari et al., PNAS 2018). In this context, we have characterized genes (*Rev7*, *53bp1*) for their functions during CSR with gene-deletion CH12F3 lines or mouse models, and these two systems always yield same conclusions (including end resection, orientation-specific joining, etc.). Combining data from gene deletion CH12 cell lines and conditional-knockout B cells, we believe the current evidence strongly support our conclusions. In addition, as suggested by Reviewer 1, we also edit the text to make our statement much clearer.

3- Most of the mutation data is generated from loci which are tightly linked to the negative outcomes of Rev7 deficiency after CSR (i.e. 5' Sμ where long resections can delete this region). One potential way to assay for SHM in B cells that haven't undergone CSR is to sort for IgM expressing B cells and look for V region mutation load and the associated pattern. Sequencing both productive and non productive rearrangements linked with unswitched alleles (e.g. RNA seq) could also help generate useful data to support SHM-specific mutation outcomes.

Response 3: We appreciate this kind suggestion. As discussed above, SHM and CSR are highly linked processes and the dissection of SHM in REV7 deficiency is complicated by its role in CSR. In this context, we access the SHM-specific mutagenic outcome of REV7 deficiency with different approaches, including the SHM analysis of Jκ5 intron and the experiments proposed by the reviewer. We are confident about our conclusions as explained in the following:

1) We have examined the SHM of Jκ5 intron, at which locus SHM happens independent and is not affected by the *IgH* CSR. We found the mutation frequency is significantly decreased in REV7 deficient cells. The phenotype can be attributed to REV7's functions in TLS to bypass AP sites and function as the main TLS extender.

Rebuttal Fig. 2 SHM analysis of J_H4 and $J_{\kappa}5$ introns in IgM^+ splenic GC B cells

a Mutation frequency ratio (*Rev7 KO* vs WT) of each single nucleotide in J_H4 intron and $J_{\kappa}5$ intron is plotted as Tukey's box plot. Nucleotides, with mutation frequency higher than 0.001, are included in the analysis. One sample *t*-test are performed with theoretical mean defined as 1. Mutation spectrums of indicated genotype in J_H4 intron (**b**) and $J_{\kappa}5$ intron (**c**), with percentage of total mutations showed as heatmaps. Percentages of specific mutation type in total mutations of J_H4 intron (**e**) and $J_{\kappa}5$ intron (**f**) regions are plotted. Two-tail unpaired *t*-test was performed for Panel **e** and **f**. ****: $p < 0.0001$, ***: $p < 0.001$, **: $p < 0.01$, *: $p < 0.05$.

2) As kindly suggested by this reviewer, the IgM⁺PNA^{hi} splenic GC B cells were sorted from *Cd19creRev7^{fl/fl}* and *Cd19cre* mice. The J_H4/J_K5 intron was amplified and subjected to SHM analysis (Rebuttal Fig. 2). We found that the mutation frequency at those regions is decreased in REV7 deficient cells (Rebuttal Fig. 2a), and a mild decreased C/G transversion was also observed (Rebuttal Fig. 2c). However, we also notice that there are a few caveats in this approach, e.g. the approach cannot distinguish the productive and non-productive V(D)J alleles, and the intra-S_μ region deletion could also complicate the SHM analysis.

3) RNA-seq data could help to distinguish the productive and non-productive alleles, which could be an alternative to bypass the above-mentioned caveats. Recently, several methods have been developed to retrieve BCR sequence from RNA-seq data including MiXCR (Bolotin et al., Nat Biotechnol 2017), TRUST (Hu et al., Nat Genet 2019), etc. Among those algorithms, only TRUST was applied to access SHM of *IgV* CDR3 with RNA-Seq data. As suggested by this reviewer, we performed RNA-seq of IgM⁺PNA^{hi} splenic GC B cells with difference genotypes (*Cd19cre* and *Cd19creRev7^{fl/fl}*). Although we have retrieved thousands of *IgH* CDR3 sequences from our RNA-seq data, the SHM cannot be easily assayed. We looked into the protocol details (Hu et al., Nat Genet 2019), and realized that the SHM was estimated by mutations within a BCR cell cluster and mutation direction was assigned with CDR3 pairs from difference *Ig* class (Hu et al., Nat Genet 2019). Thus, in our current IgM⁺PNA^{hi} splenic GC B cells RNA-seq data, the SHM analysis is not doable.

4) We also examined the SHM frequency at *Ig V* introns in *53bp1^{-/-}* GC B cells. Although 53BP1 deficiency results in the most severe expanded resection of S region breaks during CSR comparing to the other gene knockouts in the 53BP1-RIF1-Shieldin pathway (Fig. 2c), it does not affect SHM frequency at *Ig V* introns (Supplementary Fig. 11). The result suggests that the expanded S region resection during CSR does not affect the *Ig V* introns mutation during SHM.

In sum, we have used different approaches to assay the SHM-specific mutation outcomes in REV7 deficiency, and conclude that REV7 is required for normal SHM.

4- The G2/M phase arrest should be better described in CH12 cells as this does not appear to be AID-dependent.

Response 4: We thank this reviewer for her/his appreciation of the experiments. We performed new experiments in CH12F3 cell lines (Supplementary Fig. 7) and also further discuss the G2/M phase arrest in REV7 deficiency.

As this reviewer pointed out, the significant G2/M phase arrest in REV7 deficiency is not AID (or DNA damage)-dependent. For example: *Aicda* knockout cannot rescue the G2/M phase arrest in *Cd19creRev7^{fl/fl}* CSR-activated B cells (Fig. 4e); when *Aicda* is not expressed under non-cytokines culture condition, significant G2/M phase arrest was observed in REV7 deficiency (Supplementary Fig. 7). The G2/M phase arrest could be a direct result of failed spindle assembly (Bhat et al., Cell Cycle 2015).

In the revision, we carefully performed new experiments and checked cell cycle distribution of different CH12F3 cell lines with or without cytokine-activation (Supplementary Fig. 7). Similar to previous observation (Di Virgilio et al., Science 2013), RIF1 deficiency results to significant of G2/M arrest. Similar G2/M arrest phenotype was observed in both

REV7 and REV3L deficiencies regardless of cytokine-stimulation. We also noticed a slightly increased G2/M arrest (not statistically significant in the current experiment setting) in *53bp1*^{-/-} CSR-activated B cells (Fig. 1f) and CH12F3 cell lines (Fig. 7b), reflecting a general DNA damage response.

5- Cell death is linked to AID-deamination which leads to CSR and SHM (in vivo). The relative contribution of each process is unclear when both are activated as is the case in GC's. This is not clear and should be dissected. Looking at unswitched B cells for SHM and having markers for Rev7 deletion would help this analysis.

Response 5: We thank this reviewer for her/his interest into this topic. Although we conclude the cell death in REV7-deficiency can be contributed by both SHM and CSR, further dissection of their contribution in GC B could be help to understand the differential outcomes. As mentioned above in the Response 3 to this reviewer, we have checked the mutation profiles of J κ 5 intron in GC B cells and J μ 4/J κ 5 introns in unswitched GC B cells. We confirmed the observation of SHM-specific outcomes in REV7 deficiency. We describe these results in the revised text and also include some data in this rebuttal letter (Rebuttal Fig. 2). We also agree that having a marker for *Rev7* deletion will strengthen the conclusion, which could be achieved by designing a new KO strategy (e.g. a “knock-out first” strategy or a conditional inversion strategy) and cannot be finished in a short time. It is of note that a recent study (Roco et al. Immunity 2019) claimed that CSR infrequently happens in germinal centers. Although the conclusion needs to be further confirmed by others in the field, it suggests that the cell death of GC B cells could be mainly contributed by deleterious intermediate products during SHM in REV7 deficiency *in vivo*.

6- The use of CRISPR/Cas9-generated breaks at the Ig3 region of CH12 cells is unclear and should be better justified.

Response 6: We thank this reviewer for giving us a chance to explain the experiment design.

The CRISPR/Cas9 break at Ig3 in CH12 cells is used as a bait break to capture the gene rearrangements in *IgH* locus in HTGTS assay. The reason is that in CH12F3 cells one *IgH* allele already switched. Here is a schematic illustration of *IgH* locus in CH12F3 cells, primary naïve B cells and germ line (Rebuttal Fig. 3).

Rebuttal Fig. 3: Schematic illustration of *IgH* locus in different cells.

IgH locus is illustrated for germline, CSR-activated B cells and CH12F3 cell line. HTGTS primer sites are marked with blue arrows.

In HTGTS assay, translocation is cloned from a bait break, and the translocation junction analysis reveals the end-joining between bait break and prey breaks. In CSR-activated primary B cells, the *IgH* constant genes in both alleles are in germline configuration. A 5'S μ primer was applied to clone the S-S junctions (Dong et al., Nature 2015). However, in the CH12F3 cells, the nonproductive DJ allele has already undergone CSR to IgA. The same 5'S μ primer cannot be used to clone *IgH* junctions as the S μ -S α in nonproductive DJ allele will interfere the downstream analysis. Thus, we introduce a CRISPR/Cas9 break at I γ 3 region, which can capture AID-initiated breaks on the productive VDJ allele during CSR from IgM to IgA. Designer endonucleases, including I-Sce1 and Cas9, are the most frequently used tools to generate bait breaks in HTGTS technology (Chiarie et al., Cell 2011; Frock et al., Nat Biotechnol 2015). In this context, the I γ 3^{Cas9}-S α ^{AID} junction analysis have yielded same conclusions as the S μ ^{AID}-S α ^{AID} junction analysis, including expanded end resection in *53bp1*^{-/-} B cells, orientation-specific CSR end-joining, etc. We put the above description of HTGTS assay in the Method section of the revised manuscript.

7- The base editing data are of some interest but are largely undeveloped and speculative, and outside the focus of the study.

Response 7: We agree with this comment. We move the panels to Supplementary Figures, and keep the discussion minimal.

Reviewers' comments:

Reviewer #1 (Remarks to the Author):

The authors have satisfactorily addressed all the issues raised by this reviewer on the first submission. Furthermore, they have rearranged the manuscript to present their findings in a more structured and clearer way. I am happy to recommend it for publication in Nature Communications.

Michela Di Virgilio

Reviewer #2 (Remarks to the Author):

The authors have satisfactorily addressed all the comments raised by this reviewer.

Reviewer #3 (Remarks to the Author):

The revised manuscript submitted from Yang et al. examines Rev7 in programmed DNA breaks in mature B cells. They assay for outcomes of AID-mediated genetic modifications in B cells undergoing CSR and/or SHM in a mouse model with lineage specific inactivation of Rev7 or in cells lines engineered with mutations in Rev7 (and associated partners). They find that Rev 7 is necessary for CSR and for cell survival after switching, independent of its role in DSBR or G2/M transition. When AID-dependent SHM is assessed, the overall frequency is diminished in Rev7 deficiency however there is a difference in the spectrum of sequence mutation between the V regions and the IgH S regions. They detect increased cell death after CSR stimulation which this group attributes to AID activity, rather than Rev7-dependent G2-M arrest. Additional experiments with mutational studies of the residues in HORMA domain of Rev 7 demonstrate that it is crucial for B cell survival. There are several aspects of the study that require attention to better understand the Rev7 activities that contribute to cell death.

Comments

1- The abrupt cell death seen in Rev7-deficient between day 3 and day 4 is puzzling, as CSR occurs even at Day 3 (seen in Supplemental fig 1). Moreover, apoptosis for LPS plus IL4 stimulated B cells appears similar to controls (S1J). This should be reconciled as it is unclear whether this is due to the relatively few experimental points (4) or whether this is due to cell death or cells arrested at G2/M. While there can be loss of cells outside of apoptotic pathways, it is unclear as to how different stimulations can cause this effect.

2- The 5'S_μ SHM assay reveals that Rev7 deficiency has fewer mutations after CSR (SF4) and loss of C>G fraction (F2d). It is difficult to assess the significance of the percentage decrease of the C>G fraction without seeing the global changes in other sequences in this region. This should be clearly depicted as to better understand the mechanism whether it be deletion or changes in TLS.

3- Figure panel 4E shows that there are more cells in G2/M arrest after LPS + IL4 stimulation, yet this is interpreted as not a cause of cell loss after AID lesions are generated. This is not well discussed, and it would be useful to show the cell cycle data in AID proficient cells.

4- SHM at the VH or VK regions appear to be reduced with no change in spectrum, in contrast to the sequence alterations seen in the 5'S_μ locus. This is not completely discussed for the reader to understand the sequence specificity (versus locus) of this finding.

5- The discussion should be expanded to better understand the outcomes of AID activity in Rev7

deficient cells.

#NCOMMS-19-23548B

Response to Reviewer's comments (original reviewers' comments are in italic):

Reviewers' comments:

Reviewer #1 (Remarks to the Author):

The authors have satisfactorily addressed all the issues raised by this reviewer on the first submission. Furthermore, they have rearranged the manuscript to present their findings in a more structured and clearer way. I am happy to recommend it for publication in Nature Communications.

Michela Di Virgilio

Response: We thank Dr. Di Virgilio for her positive and prompt evaluation of our revised manuscript.

Reviewer #2 (Remarks to the Author):

The authors have satisfactorily addressed all the comments raised by this reviewer.

Response: We thank Reviewer 2 for her/his positive and prompt evaluation of our revised manuscript.

Reviewer #3 (Remarks to the Author):

The revised manuscript submitted from Yang et al. examines Rev7 in programmed DNA breaks in mature B cells. They assay for outcomes of AID-mediated genetic modifications in B cells undergoing CSR and/or SHM in a mouse model with lineage specific inactivation of Rev7 or in cells lines engineered with mutations in Rev7 (and associated partners). They find that Rev 7 is necessary for CSR and for cell survival after switching, independent of its role in DSBR or G2/M transition. When AID-dependent SHM is assessed, the overall frequency is diminished in Rev7 deficiency however there is a difference in the spectrum of sequence mutation between the V regions and the IgH S regions. They detect increased cell death after CSR stimulation which this group attributes to AID activity, rather than Rev7-dependent G2-M arrest. Additional experiments with mutational studies of the residues in HORMA domain of Rev 7 demonstrate that it is crucial for B cell survival. There are several aspects of the study that require attention to better understand the Rev7 activities that contribute to cell death.

Response: We thank this reviewer for the helpful comments and will address them point-by-point as following.

Comments

1- The abrupt cell death seen in Rev7-deficient between day 3 and day 4 is puzzling, as CSR occurs even at Day 3 (seen in Supplemental fig 1). Moreover, apoptosis for LPS plus IL4 stimulated B cells appears similar to controls (S1J). This should be reconciled as it is unclear whether this is due to the relatively few experimental points (4) or whether this is due to cell death or cells arrested at G2/M. While there can be loss of cells outside of apoptotic pathways, it is unclear as to how different stimulations can cause this effect.

Response 1: We thank this reviewer for the comment. To clarify these two unclear points, we now present more data in CSR-activated B cells upon LPS/IL4 or LPS stimulation at both Day 3 and 4. Here, we show a side-by-side comparison including CSR level, cell number count, apoptosis, live cell population gated on flowcytometry forward/side scatter plot, and cell cycle as Rebuttal Figure 1. Data which are added during this revision are highlighted (Rebuttal Fig. 1c, d, e, f Day 3 data; g, h and j).

1.1 Abrupt cell death between Day 3 and 4: Although substantial CSR levels were observed at Day 3 in our current B cell culture condition, more CSR events happened between timepoints at Day 3 and 4 (Rebuttal Fig. 1a, b). As an example, CSR to IgG1 of LPS/IL4 stimulated wild type B cells increased from ~25% to ~55% between Day 3 and 4. Accordingly, we detected significant cell death of REV7 deficient B cells at Day4 timepoint, as we conclude “Dramatic growth defect of REV7 deficient B cells was observed at Day 4 after cytokine stimulation”. When analyzing the cell number (Rebuttal Fig. 1c, d), apoptosis (Rebuttal Fig. 1e, f) and live cell population gated on forward/side scatter plot (Rebuttal Fig. 1g, h), we also noticed less dramatic cell death at Day 3 comparing to the level at Day 4 timepoint (Rebuttal Fig. 1c-h, comparing data at Day 3 and Day 4). Thus, we conclude that the extensive CSR events happened between Day 3 and 4 timepoints contribute to the much severe cell death we observed at Day 4 in our current culture condition.

1.2 Difference of apoptosis upon LPS/IL4 or LPS stimulation: With side-by-side comparison of different phenotypes upon LPS/IL4 or LPS stimulation, we found the cell number count (Rebuttal Fig. 1c versus d), live cell population gated on forward/side scatter plot (Rebuttal Fig. 1g versus h), and cell cycle (Rebuttal Fig. 1i versus j) show similar trends. Only the apoptotic fraction upon LPS/IL4 stimulation is less dramatically changed in REV7 deficient B cells (Rebuttal Fig. 1e). Upon LPS/IL4 stimulation, B cells switch from IgM to IgG1 and IgE, while upon LPS-alone stimulation B cells switch from IgM to IgG3 as different stimulation turns on the corresponding *Ig* constant gene germline transcription (Stavnezer et al., *Annu Rev Immunol*, 2008). In our current culture condition, we noticed that cell proliferated slower (Rebuttal Fig. 1c versus d, different Y axis) and slightly less live cell population gated on forward/side scatter plot in presence of IL4 (Rebuttal Fig. 1g versus h, Day 4). IL4 initiates several signal transduction pathways in B cells including an anti-apoptotic pathway (Wurster et al., *JBC*, 2002). There could be loss of cells outside of apoptotic pathway upon LPS/IL4 stimulation as this reviewer suggested. In the cell death data obtained from either cell number counting or live cell population gated on forward/side scatter plot, REV7 deficiency always leads to severe cell loss upon AID expression. Thus, the difference of LPS/IL4 or LPS stimulation did not affect our main conclusions.

Accordingly, we re-arranged Figure 1 and Supplementary Figure 1 and 3 to include the new data, and also re-edit the corresponding result section (Page 5, Line 122-124; Page 6, Line 128-132). We thank this reviewer for these suggestions to make our interpretation much clear.

Rebuttal Fig 1. CSR and cell death of activated B cells upon LPS/IL4 or LPS stimulation at different timepoints. (New data and analyses are highlighted)

- a. CSR levels to IgG1 after LPS/IL4 stimulation at Day 3 and 4.
- b. CSR levels to IgG3 after LPS stimulation at Day 3 and 4.
- c. Cell numbers of activated B cells after LPS/IL4 stimulation at Day 3 and 4.
- d. Cell numbers of activated B cells after LPS stimulation at Day 3 and 4
- e. Percentages of apoptotic cell after LPS/IL4 stimulation at Day 3 and 4.
- f. Percentages of apoptotic cell after LPS stimulation at Day 3 and 4.
- g. Live cell population gated on forward/side scatter plot after LPS/IL4 stimulation at Day 3 and 4.
- h. Live cell population gated on forward/side scatter plot after LPS stimulation at Day 3 and 4.
- i. Percentage of cells in G2/M phase is plotted after LPS/IL4 stimulation at Day 3.
- j. Percentage of cells in G2/M phase is plotted after LPS stimulation at Day 3.

Data are represented as mean \pm SD in all panels. One-way ANOVA followed by Dunnett's multiple comparisons test was performed for all panels. Data from *Rev7* knockout are compared with those from other genotypes. ****: $p < 0.0001$, ***: $p < 0.001$, **: $p < 0.01$, *: $p < 0.05$, ns: $p > 0.05$.

2- The 5'S μ SHM assay reveals that Rev7 deficiency has fewer mutations after CSR (SF4) and loss of C>G fraction (F2d). It is difficult to assess the significance of the percentage decrease of the C>G fraction without seeing the global changes in other sequences in this region. This should be clearly depicted as to better understand the mechanism whether it be deletion or changes in TLS.

Response 2: We thank this reviewer for this comment. We have performed the suggested experiments and present the data here.

In brief, we amplified the VDJ exon sequence and a control sequence on Chromosome 12 in CH12F3 cells (Rebuttal Fig. 2a top). The control sequence (chr12:25651533- 25651774) locates at an un-transcribed intergenic region, which is unlikely an AID target as transcription is required for AID targeting. We checked the mutation profile of this control region on chr12, and found extremely low mutation frequency (<0.05% mutation rate, i.e. <10 substitution events when the sequencing depth is 20000 reads per nucleotide). The background mutation could be introduced by rare random mutagenesis, PCR amplification, sequencing etc. By contrast, the VDJ exon underwent AID-initiated mutagenesis as most of the mutations located in WRC motifs (Rebuttal Fig. 2a). In this context, the "AGCT" motif inside CDR3 stood up to be the most significant hotspot (Rebuttal Fig. 2a). However, the VDJ exon underwent AID-dependent mutagenesis at a 5-fold lower frequency comparing to the 5'S μ sequence (Rebuttal Fig. 2a, different Y axis).

To exclude the background mutation, we focused on the mutation analysis of the "AGCT" hotspot inside CDR3. REV7 deficiency leads to a decreased mutation frequency at the AGCT hotspot (Rebuttal Fig. 2b). In total, there were 671 substitution events in wildtype CH12F3 cells and 400 substitution events in REV7 deficient CH12F3 cells from four different repeats. We found that in wild type CH12F3 cells, C>G transversion only takes ~2% of the total substitution events at the AGCT hotspot inside CDR3 (comparing to ~15% at 5'S μ region), and REV7 deficiency did not significantly change the mutation spectrum (Rebuttal Fig. 2c). There are three possible reasons: 1), the mutation frequency is too low to allow the mutation spectrum analysis. For example, total 16 C>G transversion events was obtained in wildtype CH12F3 (at the "AGCT" hotspot, total 299115 counts at G/C, 671 substitution at G/C and 16 G>C/C>G transversion). 2). Differential downstream DNA repair pathways function at VDJ exon and the 5'S μ sequence in CH12F3 cells, as similar phenomenon has discovered in germinal center B cells (Liu et al., *Nature* 2008). In this context, the two sequences in the *IgH* locus of CH12F3 cells could be a tool to study this long-sought question. 3) It's of note that the mutation profile including frequency and spectrum of VDJ exon in CH12F3 cells was similar to the profiles of *Ig V* introns in GC B cell, suggesting a sequence intrinsic feature might contribute to the different mutation profiles of V sequence and S sequence.

When comparing the S region resection and mutation spectrum (Figure 2c and d), we noticed that extended S region resection was observed in cells deficient for 53BP1-RIF1-Shieldin (Figure 2c), while the decreased C>G transversion was observed in cells deficient for REV1/3L/7 (Figure 2d, last seven bars). However, we also observed less severe decreased C>G transversion in 53BP1, RIF1, SHLD3 deficiencies (Figure 2d, comparing the first wildtype and the second to fourth bars). Thus, we conclude that sequence deletion due to extended S region resection could contribute to decreased C>G transversion, while the absence of TLS is the major cause of decreased C>G transversion in REV7 deficiency.

In the revised manuscript, we insert the new data as revised Supplementary Figure 12 and also discuss the implications in result (Page 8, Line 199-202) and discussion section (Page 12, Line 308-310) as marked in red.

Rebuttal Fig 2. Mutation profiles of three sequences on chr12 in CH12F3 cells.
a Mutation frequency of three sequences on chromosome 12 are plotted. Location and annotation are illustrated on top. Mutation profiles are plotted as Supplementary Fig. 4a. Complementarity-determining regions (CDR) are bracketed with dashed lines. The AGCT hotspot inside CDR3 is marked by a red triangle.
b Mutation frequency of C/G at AGCT hotspot inside CDR3.
c Mutation spectrum of C/G of the AGCT hotspot inside CDR3.
 Data are represented as mean \pm SD in Panel b and c. Two-tail unpaired t-test was performed for Panel b and c. *: $p < 0.05$, ns: $p > 0.05$.

3- Figure panel 4E shows that there are more cells in G2/M arrest after LPS + IL4 stimulation, yet this is interpreted as not a cause of cell loss after AID lesions are generated. This is not well discussed, and it would be useful to show the cell cycle data in AID proficient cells.

Response 3: We are sorry that we did not make the point clear. We have showed cell cycle data in AID proficient cell in Figure 1f and also add additional new data in the revised Supplementary Figure 3. Those data were obtained under same culture condition, and the levels are comparable. We add more details in the result section (Page 10, Line 254-256) and also discuss more in the discussion section (Page 13, Line 327-329).

4- SHM at the VH or VK regions appear to be reduced with no change in spectrum, in contrast to the sequence alterations seen in the 5'S μ locus. This is not completely discussed for the reader to understand the sequence specificity (versus locus) of this finding.

Response 4: We thank this reviewer for giving us a chance to further discuss this point. We added detailed discussion (Page 12-13, Line 310-325) to cover the SHM

frequency/spectrum, and discussed the possible contribution of intrinsic sequence difference and potential different DNA repair pathways.

5- The discussion should be expanded to better understand the outcomes of AID activity in Rev7 deficient cells.

Response 5: We rewrite the previous second paragraph of the discussion section and further discuss the different outcomes of AID activity in REV7 deficient cells (Page 12-13). We thank this reviewer for giving us a chance to better discuss our data.

REVIEWERS' COMMENTS:

Reviewer #2 (Remarks to the Author):

All points have been addressed. This is a nice study.

#NCOMMS-19-23548B

Response to Reviewer's comment (original comment are in italic):

REVIEWERS' COMMENTS:

Reviewer #2 (Remarks to the Author):

All points have been addressed. This is a nice study.

Response: We thanks this reviewer for her/his positive evaluation of our revised manuscript. We are delighted that this reviewer think it is a nice study.